# Biological heterogeneity in idiopathic pulmonary arterial hypertension identified through unsupervised transcriptomic profiling of whole blood

Sokratis Kariotis [1,2], Emmanuel Jammeh[1,2], Emilia M. Swietlik[3,4], Josephine A. Pickworth [2], Christopher J. Rhodes [5], Pablo Otero[5], John Wharton [5], James Iremonger [2], Mark J. Dunning [1], Divya Pandya[3], Thomas S. Mascarenhas[1], Niamh Errington [1,2], A. A. Roger Thompson [2,6], Casey E. Romanoski[7], Franz Rischard[7], Joe G. N. Garcia[7], Jason X.-J. Yuan[8], Tae-Hwi Schwantes An[9], Ankit A. Desai[9], Gerry Coghlan[10], Jim Lordan[11], Paul A. Corris[11], Luke S. Howard [5], Robin Condliffe[2,6], David G. Kiely[2,6,12], Colin Church[13], Joanna Pepke-Zaba[4], Mark Toshner[3,4], Stephen Wort[5], Stefan Gräf [3], Nicholas W. Morrell[3], Martin R. Wilkins [5], Allan Lawrie [2,12,30 ✉], Dennis Wang [1,14,15,30 ✉] & UK National PAH Cohort Study Consortium*

Idiopathic pulmonary arterial hypertension (IPAH) is a rare but fatal disease diagnosed by right heart catheterisation and the exclusion of other forms of pulmonary arterial hypertension, producing a heterogeneous population with varied treatment response. Here we show unsupervised machine learning identification of three major patient subgroups that account for 92% of the cohort, each with unique whole blood transcriptomic and clinical feature signatures. These subgroups are associated with poor, moderate, and good prognosis. The poor prognosis subgroup is associated with upregulation of the *ALAS2* and downregulation of several immunoglobulin genes, while the good prognosis subgroup is defined by upregulation of the bone morphogenetic protein signalling regulator *NOG*, and the C/C variant of *HLA-DPA1/DPB1* (independently associated with survival). These findings independently validated provide evidence for the existence of 3 major subgroups (endophenotypes) within the IPAH classification, could improve risk stratification and provide molecular insights into the pathogenesis of IPAH.

[1] Department of Neuroscience, University of Sheffield, Sheffield, UK. [2] Department of Infection, Immunity & Cardiovascular Disease, University of Sheffield, Sheffield, UK. [3] Department of Medicine, University of Cambridge, Cambridge, UK. [4] Royal Papworth Hospital, Cambridge, UK. [5] National Heart and Lung Institute, Imperial College London, London, UK. [6] Sheffield Pulmonary Vascular Disease Unit, Royal Hallamshire Hospital, Sheffield, UK. [7] Department of Cellular and Molecular Medicine, University of Arizona, Tucson, AZ, USA. [8] Department of Medicine, University of California, San Diego, La Jolla, CA, USA. [9] Department of Medicine, Indiana University, Indianapolis, IN, USA. [10] Royal Free Hospital, University College London, London, UK. [11] Newcastle University, Newcastle, UK. [12] Insigneo institute for In Silico Medicine, Sheffield, UK. [13] University of Glasgow, Glasgow, UK. [14] Department of Computer Science, University of Sheffield, Sheffield, UK. [15] Singapore Institute for Clinical Sciences, Singapore, Singapore. [30] These authors jointly supervised this work: Allan Lawrie, Dennis Wang. *A list of authors and their affiliations appears at the end of the paper. ✉email: a.lawrie@sheffield.ac.uk; dennis.wang@sheffield.ac.uk

Pulmonary arterial hypertension (PAH) is a rare but devastating disease characterised by sustained pulmonary vasoconstriction and progressive pulmonary vascular remodelling. This leads to an increase in pulmonary vascular resistance and pulmonary artery pressure, resulting in right heart failure and death[1]. The cause of idiopathic PAH (IPAH) remains unknown and diagnosis is derived from the exclusion of other forms of PAH, resulting in a heterogeneous group of patients who have significant differences in survival and treatment response across clinical cohort and registry studies[2–5].

The pathobiology of PAH involves the complex interaction of resident vascular cells, including endothelial cells, arterial smooth muscle cells and fibroblasts, with infiltrating inflammatory cells, and has been shown to be regulated by an ever growing number of molecular and genetic mechanisms[6–8]. We have identified both rare mutations[9] and common variants[10] in heritable and idiopathic PAH (H/IPAH) that have provided further insight into the genetic underpinning of PAH. Additional proteomic[11], metabolomic[12] and transcriptomic[13] studies have described diagnostic and prognostic biomarkers that add to our increasing understanding of the molecular mechanisms that regulate disease in this cohort. In Rhodes et al. we compared clinically defined H/IPAH cases to healthy controls and defined an imperfect diagnostic signature for H/IPAH; however, we have not previously examined the molecular heterogeneity that exists within H/IPAH cases. Deep RNA profiling of blood samples have provided accessible biomarkers to detect rare diseases[14] and defined molecular mechanisms behind myocardial infarction[15]. We therefore investigated whether transcriptomic profiling of whole blood can provide more granular molecular 'endophenotypes' of H/IPAH to stratify patients better than is currently permissible with the standard clinical classification. Furthermore, we hypothesised that these transcriptome-defined subgroups would provide additional insights into biological pathways driving disease, and potential drug targets offering a route to precision medicine approaches for H/IPAH.

In this study, assessment of transcriptome patterns in whole blood was conducted using unsupervised machine learning agnostic to the clinical definitions and descriptors of H/IPAH. We describe the unbiased partitioning of patients into multiple distinct transcriptomic subgroups that associate with different survival properties, each with predictive clinical and genetic features. Specifically, we highlight the potential role of immunity and immune genes in discrimination of PAH endophenotypes associated with differential patient outcomes. These data further highlight the concept that inflammation is an important mediator of PAH pathogenesis[16–22] and the discovery of distinct immune subgroups from blood cytokine profiles of patients with PAH[16–18]. Finally, we identify a specific panel of clinical features that describe each transcriptomic subgroup and replicate these subgroups in a validation cohort who did not undergo full transcriptomic profiling using their clinical phenotype data. The gene expression profile of key cluster associated genes was subsequently confirmed, and the correlation with key clinical variables validated in both internal and external validation cohorts, thereby validating our approach, and providing an alternative method to define these endophenotypes without the need for transcriptomic data.

## Results

**Unsupervised cluster analysis of whole-blood transcriptomes reveals five distinct subgroups of H/IPAH.** Whole blood samples from patients with H/IPAH ($n = 359$) were processed for RNA-sequencing as previously described[13]. Samples from 359 patients and 21 samples collected from a second time point

underwent RNAseq data processing to reduce noise, and gene filtering to remove gender bias as sex chromosomes produced the highest variation in gene expression during clustering (Supplementary Fig. 1). Sample collection site did not produce any discernible effect on clustering (Supplementary Figs. 2 and 3). Simultaneously, the 300 genes that produced the most stable expression dataset were utilised to identify unique subgroups of gene expression profiles and describe the biological and clinical descriptors of these subgroups (Fig. 1). A clustering algorithm for selection and majority voting of multiple internal validation indexes (Supplementary Data 1) allowed us to identify as statistically optimal five distinct and stable subgroups of patients' profiles (Fig. 2a) while retaining the maximum heterogeneity information found in our dataset. The largest of the patient clusters identified was subgroup I ($n = 129$), which had poorer survival (53%, five-year median survival from sampling; Fig. 2b). The second largest, subgroup II ($n = 112$), demonstrated the best survival (78%, 5 years from sampling; Fig. 2b). Subgroup V ($n = 89$) demonstrated a mixed gene expression pattern and average survival outcome compared to subgroups I and II (Fig. 2a, b). Subgroups III ($n = 19$) and IV ($n = 10$) also demonstrated distinct gene expression patterns, with subgroup III most similar to subgroup II, and subgroup IV similar to subgroup I both in terms of gene expression level and survival outcomes. Due to the small size of subgroups III and IV (making statistical significance unattainable), we focused further characterisation of

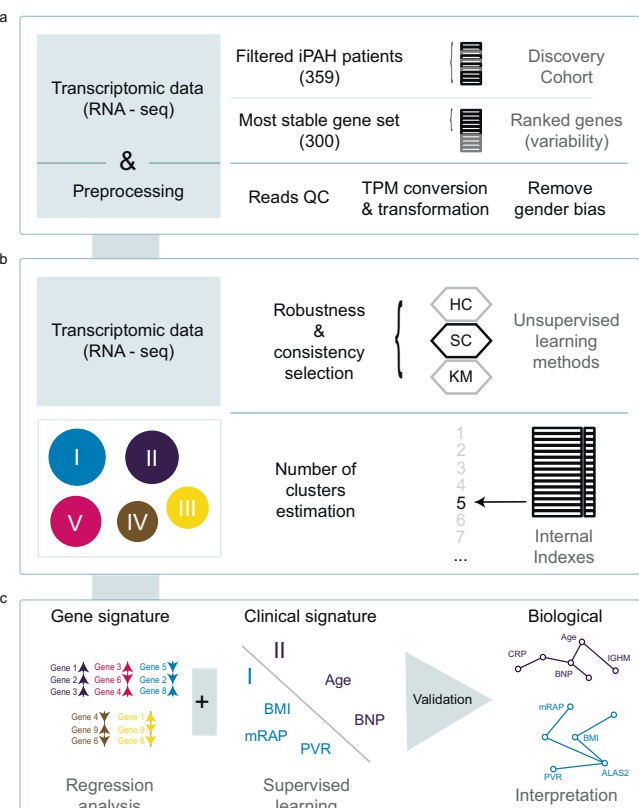

**Fig. 1 Overview of IPAH subgroup identification methodology. a** A cohort of 359 IPAH patients and a set of 300 genes are selected for clustering based on RNA data quality and variability of expression across samples. **b** Spectral clustering of patients using expression values (TPM) was benchmarked against hierarchical clustering (HC) and k-means clustering (KM), and the optimal number of IPAH subgroups was selected based on internal indexes. **c** Associated gene expression and clinical features were identified and validated in independent cohorts.

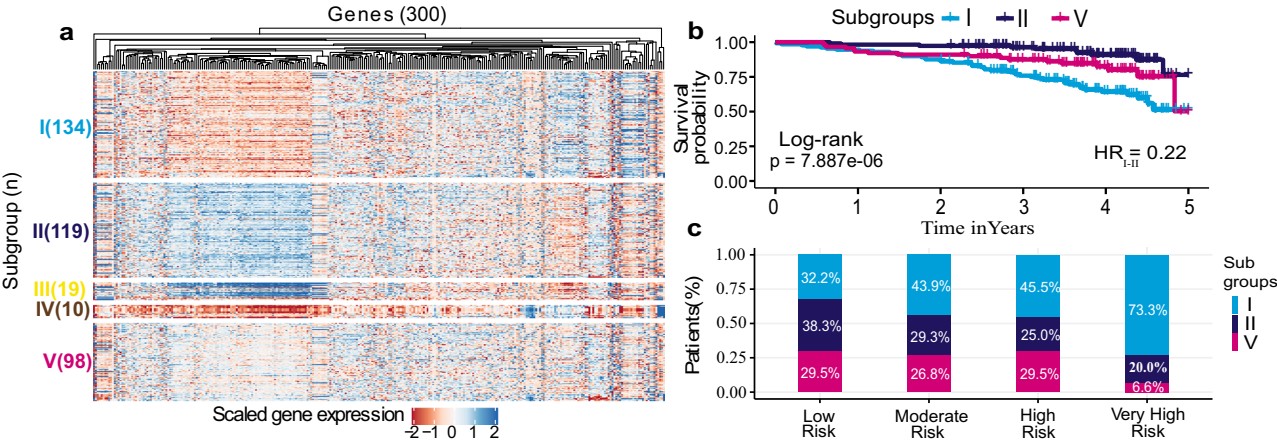

**Fig. 2 Gene expression profiles, survival and risk categories that demonstrate five distinct subgroups. a** The expression heatmap for the five discovered subgroups showing distinct expression profiles. **b** Kaplan–Meier survival curves for the three predominant subgroups demonstrating the difference in survival profiles (from RNA sampling) for a span of 5 years along with two-sided log-rank test p values. **c** The percentage of predominant subgroups I, II and V patients across REVEAL risk categories. High- and very-high-risk populations mostly consist of subgroup I patients (45.5% and 73.3%, respectively), while the low-risk population is mostly composed of subgroup II (38.3%) and V (29.5%) patients. Fisher's exact test showed a statistically significant difference (two-sided p value = 0.024) between subgroups I and II for low- and very-high-risk categories.

genetic and clinical correlates for subgroups I, II and V. The 33 HPAH patients in our PAH cohort showed an equal distribution (~10%) among the subgroups of our initial clustering (Supplementary Table 1), indicating that the inclusion of HPAH, or the small number of mis-classified patients, did not drive the partitioning procedure. An additional clustering pipeline exclusively utilising 313 samples from patients with IPAH (i.e. excluding those with HPAH, or re-classified PH) also showed five subgroups (Supplementary Fig. 4), where there were also a group of patients with poorer survival (clusters B and E, n = 149), a group with good survival (A and C, n = 109) and a group with moderate survival (D, n = 55).

In order to determine whether the survival differences between the three main (largest) transcriptomic subgroups were also associated with disease severity in the surviving patients, we calculated the REVEAL 2.0 risk score[4] across all risk levels: low (n = 146), moderate (n = 41), high (n = 44) and very high (n = 15). Subgroup I which had the worst survival also had both the highest percentage of patients in high-risk categories (medium 43.9%, high 45.5% and very high 73.3%) and a lower percentage (32.2%) in the low-risk category (Fig. 2c). In contrast, subgroup II which had the best survival was composed mostly of low-risk patients (38.3%), a proportion significantly different to subgroup I (z-test p = 0.01422, Fig. 2c). The distribution of subgroup V was uniform across the risk groups, except for a small proportion of very-high-risk patients (6.6%). Age and sex were also included as covariates with the subgroups in a Cox regression model. Age above 52 years (median) was significantly associated with poor survival (HR = 2.29) while gender showed no relationship with overall survival. Even with these covariates, subgroup I was still significantly associated with survival and was the biggest risk factor (HR = 3.83) for poor outcome (Supplementary Fig. 5). Within each subgroup, a small number of patients had a second time-point sample collected on average after 463 days. Patients with these longitudinal samples (n = 19) were found to either remain within their subgroup or transition from either subgroup I (poor prognosis) or II (good prognosis) to the moderate prognosis subgroup V (Supplementary Fig. 6a). Interestingly, no patient transitioned from subgroup II (best survival) directly to subgroup I (worst survival) or vice versa over time, 9 patients changed through the moderate subgroup, while 12 stayed in the same subgroup. Additionally, no functional class

changes observed with almost all samples belonging to functional class III. When including transcriptomes from healthy volunteers in our cluster analysis, the highest proportion of healthy volunteers (39.1%) grouped with subgroup II patients (better prognosis) (Supplementary Fig. 6b). To further investigate the defining characteristics of the three largest subgroups, we interrogated both their gene expression profiles and clinical features to define their endophenotype.

**Relative expression of immunoglobulins define RNA-based subgroups of IPAH.** We next interrogated the three largest RNA-based subgroups using a multivariate penalised regression to identify the relationship between gene expression profiles and each of the three subgroups. The most parsimonious model revealed 57 genes with measurable association to the subgroups. ALAS2 (erythroid ALA-synthase), a catalysing haeme biosynthesis enzyme, appeared in the signatures for both subgroups I and II, and was the most differentially expressed gene (>2-fold) between the two subgroups. Several immunoglobulin light chain genes (IGKV and IGLV) were key markers for the subgroups, and these were found to be either downregulated in subgroup I (poor prognosis) or upregulated in subgroup II (good prognosis; Fig. 3a). Other than immunoglobulins, Noggin, a bone morphogenetic protein 4 antagonist, and inhibitor of hypoxia-induced proliferation[23], was the gene with the highest positive regression coefficient for subgroup II, underlining its association with good prognosis. BMP antagonist Noggin and immunoglobulin genes associated with the good prognostic subgroup II were all downregulated by more than twofold in subgroup I (Fig. 3b), fitting with contemporary understanding of perturbed BMP and inflammatory signalling in PAH pathogenesis[16, 21, 24]. Across the three major subgroups, the relative expression level of immunoglobulins ranged from low, intermediate and high for subgroups I, V and II, respectively (Fig. 3a, c), while Noggin showed significantly higher expression in subgroup II (Supplementary Fig. 7).

**Differential immune cell composition between IPAH subgroups.** To ascertain whether the large expression differences in immunoglobulin genes associated with subgroups I and II also corresponded to different levels of immune activity, we deconvoluted the RNA profiles to estimate the proportions of immune

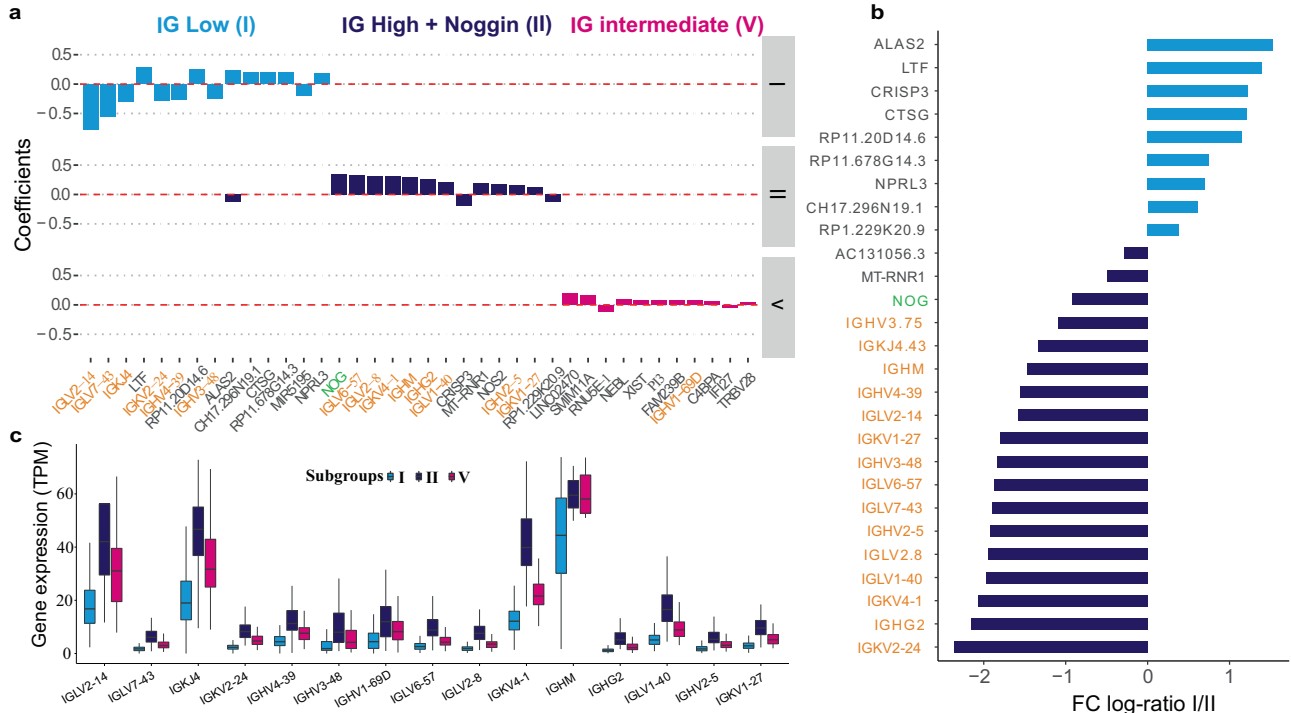

**Fig. 3 Genes associated with subgroups I (low survival), II (high survival) and V (intermediate survival). a** Genes with the highest 5% of LASSO coefficients across subgroups I, II and V. **b** Average expression fold change (log₂ scaled) of the signature genes between subgroup I and II, with significance notations. Genes over-expressed in subgroup I are denoted by light blue bars while genes primarily expressed in subgroup II are represented by dark blue bars. **c** Expression level of immunoglobulin genes selected by LASSO across the three predominant subgroups with medians shown. Subgroups I ($n = 134$), V ($n = 98$) and II ($n = 119$) can be defined as having low, intermediate and high immunoglobulin characteristics. Vertical centre line represents the median, top and bottom bounds of the box represent the first and third quartile, while the tips of the whiskers represent min and max values.

cell types in each sample. Significant differences ($p < 0.01$) in the proportion of lymphocytes and neutrophils were observed between samples in subgroup I and II (Fig. 4a and Supplementary Fig. 8). In particular, CD4/CD8 T cells and memory B cells were significantly more abundant in subgroup II where we observed upregulation of immunoglobulins. The lower proportion of lymphocytes (B cells and T cells) and higher proportion of neutrophils in the poor prognosis subgroup I was found to be statistically significant (Supplementary Table 2) and validated by clinical whole-blood cell counts (Fig. 4b). A higher neutrophil–lymphocyte ratio is known to be an indicator of poor overall survival[25]. The differences observed in CD4 T cells and memory B cells may be due to changes in MHC class II antigen presentation genes, such as *HLA-DP*. We have previously identified the *HLA-DPA1/DPB1* rs2856830 genotype to be strongly associated with survival in a large IPAH GWAS study, with the C/C homozygous genotype conferring increased survival compared with the T/T genotype, despite similar baseline disease severity[10]. Consistent with this genotype association with prognosis, we found that there was a significantly higher proportion of patients ($p = 0.009$) with the C/C genotype in subgroup II (good survival) compared with subgroup I (poor survival). This difference in variant frequencies between subgroups was not seen in known genetic risk factors for H/IPAH[9], including *BMPR2* and *SOX17* (Fig. 4c, Supplementary Fig. 9 and Supplementary Table 3).

**Common clinical characteristics across RNA subgroups.** Patients in this cohort were diagnosed at a median age of 45 years (IQR = 35–59 years) and sampled at a median age of 52 years (42–64) with an average of 5.3 years' time between diagnosis and sampling. As shown in Table 1, patients in subgroup I were significantly older ($p$ value $< 0.01$) at 57 [45–70] years than the

other subgroups. Consistent with the incidence rate of IPAH in the UK population[3], patients in the cohort were predominantly females (70%). Patients in the subgroups were also predominantly females with 62%, 73% and 70% in subgroups I, II and V, respectively. Across the whole cohort, 16.4% of patients presented positive pulmonary vasodilator response, 44.4% were in Functional Class (FC) III at sampling date with 6-minute walk distance (6MWD) of 387 m and a mean N-terminal (NT)-proBNP of 222.5 [78.9–1162.8] ng/ml. When the cohort was stratified, subgroup I had the highest proportion of FC III (50.4%), whereas subgroup II had the highest proportion of patients for FC I and II (16.5% and 41.3%, respectively, $p$ value = 0.013). The lowest 6MWD (median = 327 m, $p < 0.01$) and the highest N-terminal (NT)-proBNP was (median = 345.0 ng/ml, $p = 0.055$) were observed in patients from subgroup I (poorest survival group). Diagnostic RHC across the cohort showed mean pulmonary arterial pressure (mPAP) was 54 (46–61) mmHg, pulmonary arterial wedge pressure (PAWP) was 10 (7–12) mmHg and CO was 3.8 (3.0–4.9) l/min at diagnosis. The cohort at the time of sampling, 143 (40.2%) of the patients were FC II and 158 (44.4) FC III with a median 6MWD of 387 m, pulmonary vascular resistance (PVR) was 8.9 Wood units and an NT-proBNP 222.5 ng/ml suggestive of a slight improvement of disease phenotype in response to vasodilator therapy. The full demographics table can be found in Supplementary Data 2.

**Clinical signatures describe RNA-based subgroups.** Identification of specific clinical characteristics associated with each transcriptome-derived subgroup could explain how the gene expression patterns manifest into differences in patient outcome. We therefore used supervised machine learning with feature selection to identify the most important clinical features to

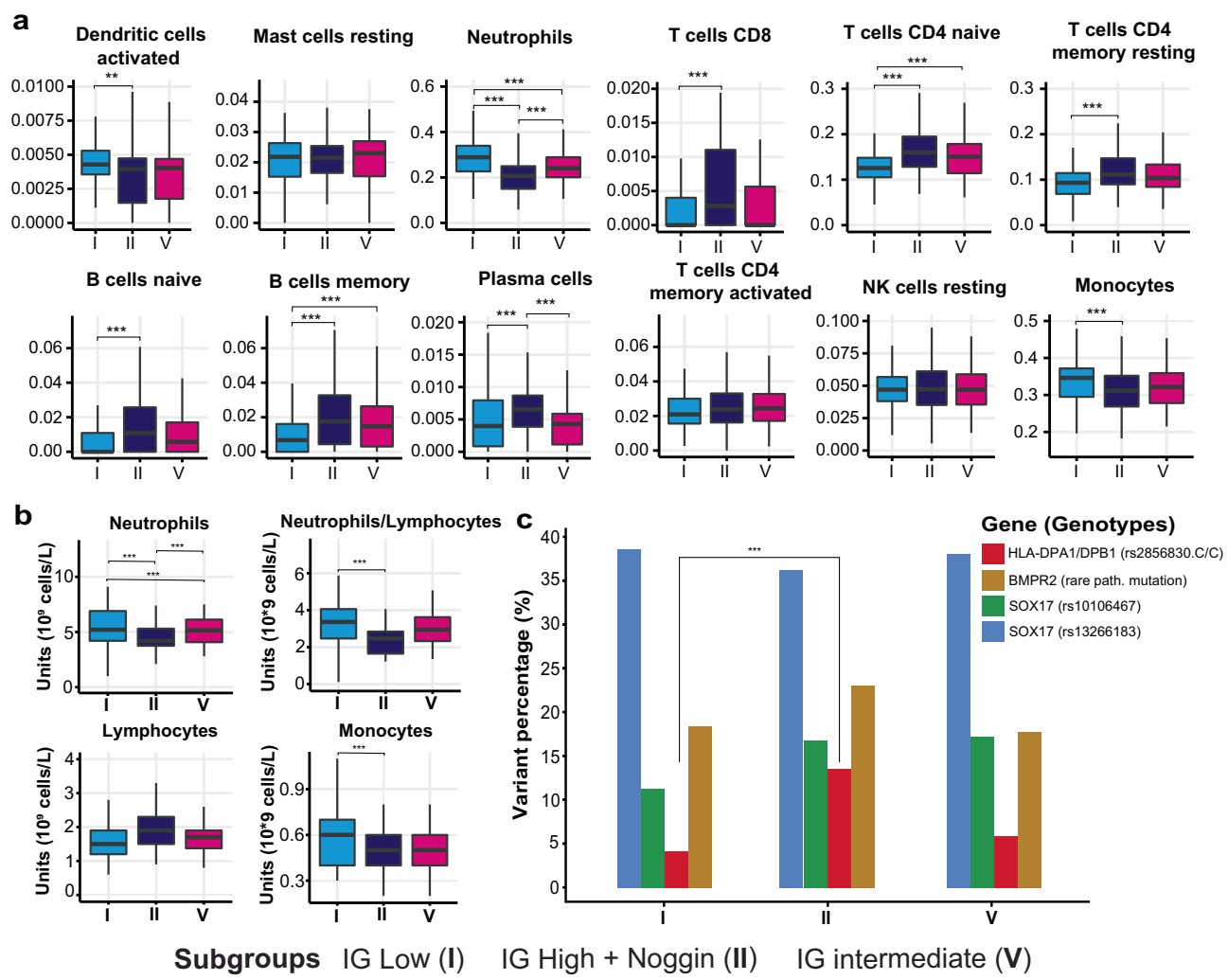

**Fig. 4 Immunity cell composition across PAH transcriptomic subgroups. a** CIBERSORT estimation of relative cell abundance in patients of subgroups I ($n = 129$), II ($n = 112$) and V ($n = 89$) using two-sided test and Bonferroni adjusted mean difference significance notation with $p$ values: $p_{I-II(Dendritic\ cells\ activated)} = 0.011$, $p_{I-II(Neutrophils)} = 4.4 \times 10^{-11}$, $p_{I-V(Neutrophils)} = 2.0 \times 10^{-3}$, $p_{II-V(Neutrophils)} = 1.7 \times 10^{-3}$, $p_{I-II(T\ cells\ CD8)} = 4.8 \times 10^{-5}$, $p_{I-II(T\ cells\ CD4\ naive)} = 1.9 \times 10^{-8}$, $p_{I-V(T\ cells\ CD4\ naive)} = 3.8 \times 10^{-3}$, $p_{I-II(T\ cells\ CD4\ memory\ resting)} = 2.3 \times 10^{-5}$, $p_{I-II(B\ cells\ naive)} = 2 \times 10^{-5}$, $p_{I-II(B\ cells\ memory)} = 2.5 \times 10^{-6}$, $p_{I-V(B\ cells\ memory)} = 3.9 \times 10^{-3}$, $p_{I-II(Plasma\ cells)} = 6.4 \times 10^{-4}$, $p_{II-V(Plasma\ cells)} = 6.5 \times 10^{-5}$ and $p_{I-II(Monocytes)} = 0.0053$. Vertical centre line represents the median, top and bottom bounds of the box represent the first and third quartile, while the tips of the whiskers represent min and max values. **b** Whole-blood cell counts across subgroups I ($n = 129$), II ($n = 112$) and V ($n = 89$) using two-sided test and Bonferroni adjusted mean difference significance notation. $p_{I-II\ (Neutrophils)} = 7.2 \times 10^{-12}$, $p_{I-V\ (Neutrophils)} = 8.0 \times 10^{-4}$, $p_{II-V\ (Neutrophils)} = 4.4 \times 10^{-4}$, $p_{I-II\ (Neutrophils/Lymphocytes)} = 0.0061$ and $p_{I-II\ (monocytes)} = 0.0076$. Vertical centre line represents the median, top and bottom bounds of the box represent the first and third quartile, while the tips of the whiskers represent min and max values. **c** Proportion of patients in each subgroup with DNA variants in *HLA-DPA1/DPB1* (rs2856830), *SOX17* (rs10106467 and rs13266183, homozygous and heterozygous), *BMPR2* (rare pathogenic variant). Notably, $p_{I-II\ (HLA-DPA1/DPB1)} = 0.009$. Generated using a two-sample test for equality of proportions with continuity correction. \**P* value ≤ 0.05, \*\**p* value ≤ 0.01, \*\*\**p* value ≤ 0.001.

describe the subgroups. The full list of clinical features used by the multivariate classifiers are described in Supplementary section 'Clinical features identification: Supervised learning' and in table format in Supplementary Data 2. Each clinical feature was assessed individually in a univariate model (Fig. 5a and Supplementary Fig. 10) and in combination with other features (multivariate model). Ensemble feature selection was used to identify reliable sets of clinical features that describe signatures for the RNA subgroups. The most important features in the signature for subgroup I irrespective of feature selection method were C-reactive protein (CRP), creatinine, age of diagnosis, body mass index (BMI) and 6MWD. For subgroup II, the important features were CRP, creatinine, age of diagnosis, BMI and 6MWD, oxygen saturation (pre-6MWD) and right atrial area (RAA) (by echocardiography). CRP, 6MWD, urate, pulmonary vascular

resistance (PVR), white blood cell count (WBC) and positive acute vasodilator challenge (at diagnostic right heart catheter) characterised subgroup IV.

CRP and 6MWD were the only clinical features present in signatures for subgroup I, II and V. Higher CRP was a marker for subgroup I, whereas lower levels indicated subgroups II and V. In contrast, 6MWD was negatively associated with subgroup I and positively with subgroups II and V. CRP showed a 37.19% increase in subgroup I compared to the average for subgroups II and V, 20.75% reduction in subgroup V compared to the average for subgroup I and II and 47.86% reduction in subgroup II compared to the average for subgroups I and V. 6MWD was 29.05% lower in subgroup I compared to the average for II and V, and increased by 7.63% in subgroup V compared to the average for II and I and 16.97% increase in subgroup II compared to the

**Table 1 Major clinical characteristics of the three main RNA subgroups in the discovery cohort (n = 359) at the time of sampling.**

|  | Low-risk subgroup II (high immunoglobulin) | Intermediate-risk subgroup V (intermediate immunoglobulin) | High-risk subgroup I (low immunoglobulin) | All patients |
|---|---|---|---|---|
| n | 112 | 89 | 129 | 359 |
| Age, years | 46 [37–56] | 52 [41–62] | 57 [45–70] | 52 [42–64] |
| Age at diagnosis, years | 41 [31–51] | 46 [37–55] | 52 [42–67] | 47 [35–59] |
| Gender:Female | 82 (73%) | 69 (78%) | 80 (62%) | 253 (70%) |
| Vasoresponse | 10 (21.7%) | 6 (13.6%) | 6 (16.2%) | 23 (16.4%) |
| **Treatments** |  |  |  |  |
| Phosphodiesterase 5 Inhibitors (PDE5i) | 12 (15.4%) | 16 (21.9%) | 22 (21.8%) | 53 (19.4%) |
| Endothelin receptor antagonist (ERA) | 6 (7.69%) | 13 (17.8%) | 8 (7.92%) | 33 (12.1%) |
| PDE5i & ERA combination | 42 (53.8%) | 30 (41.1%) | 53 (52.5%) | 134 (49.1%) |
| Prostacyclin therapy | 3 (3.85%) | 1 (1.37%) | 3 (2.97%) | 7 (2.56%) |
| Calcium channel blockers | 15 (19.2%) | 13 (17.8%) | 14 (13.9%) | 45 (16.5%) |
| **WHO functional class** |  |  |  |  |
| I | 18 (16.5%) | 10 (11.2%) | 6 (4.7%) | 35 (9.8%) |
| II | 45 (41.3%) | 36 (40.4%) | 44 (34.1%) | 143 (40.2%) |
| III | 43 (39.4%) | 40 (44.9%) | 65 (50.4%) | 158 (44.4%) |
| IV | 3 (2.8%) | 3 (3.4%) | 14 (10.9%) | 20 (5.6%) |
| 6-minute walking distance, m | 397 [338–500] | 420 [367–464] | 327 [183–390] | 387 [300–449] |
| NT-proBNP, ng/l | 131.7 [54.5–362.0] | 185.5 [76.3–463.5] | 345.0 [91.0–1556.1] | 222.5 [78.9–1162.8] |
| Forced expiratory volume [% predicted] | 92 [82–101] | 84 [72–98] | 78 [66–98] | 85 [68–100] |
| Forced vital capacity [% predicted] | 101 (20) | 99 (24) | 93 (29) | 97 (24) |
| Transfer factor of lung for carbon monoxide [% predicted] | 93 [87–106] | 97 [92–101] | 88 [67–96] | 94 [87–103] |
| **Diagnostic Right Heart Catheter Study** |  |  |  |  |
| Mean pulmonary artery pressure, mmHg | 47 [39–60] | 52 [37–65] | 56 [41–65] | 51 [39–63] |
| Mean right atrial pressure, mmHg | 8 [4–10] | 8 [4–11] | 11 [6–14] | 9 [4–12] |
| Mean pulmonary arterial wedge pressure, mmHg | 10 [7–12] | 10 [8–13] | 12 [10–14] | 11 [8–13] |
| Cardiac Index, l/min/m$^2$ | 2.3 [1.6–2.8] | 2.2 [1.7–2.4] | 1.9 [1.5–2.5] | 2.2 [1.6–2.5] |
| Pulmonary vascular resistance, Wood Units | 8.1 [5.7–14.1] | 15.0 [5.9–16.1] | 8.4 [5.9–13.2] | 8.9 [5.7–15.0] |

Intervals describe first and third quartiles. Parentheses describe standard deviation (SD).

average for I and V. Five clinical features were present in signatures for subgroup I and II but had opposite coefficients (Fig. 5b). Higher age of diagnosis, BMI, RAA and creatinine are associated in subgroup I, whereas lower levels of those three features are associated with subgroup II. Subgroup I has 17.8% higher average age compared to the average for II and V and 21.2% lower in subgroup II compared to the average for I and V. BMI was 13.1% higher in subgroup I compared to the average for II and V, and 12.9% lower in subgroup II compared to the average for I and V. Additionally, creatinine was higher by 12.8% in subgroup I compared to the average for II and V and lower by 14.4% in subgroup II compared to the average for I and V. RAA was higher by 6.8% in subgroup I and lower by 6.3% in subgroup II. In contrast, there was a 27.6% reduction of renal sodium in subgroup I compared to the average for II and V, and 26.7% increase in subgroup II compared to the average for I and V.

**Validation of clinical signatures on an independent cohort.** To validate the relationship between clinical and gene features in the RNA subgroups, we used the clinical feature signatures of the subgroups to classify patients in an independent cohort of I/HPAH patients (n = 197) where whole-blood RNA profiling was not performed (Fig. 5c). Similar to the discovery cohort, patients were diagnosed at a median age of 52 years (IQR = 39–67) and

67% were female. In all, 17.7% of the patients showed positive pulmonary vasodilator response and the majority were categorised in Functional Class III (66%) with a 6MWD of 295 m (170–396) and NT-proBNP of 796 ng/pl (128–1092). Their mPAP was 51 mmHg (42–57) and PAWP was 9 mmHg (6–11). The clinical features associated with RNA subgroups from the discovery cohort were used to classify this validation cohort. Our supervised approach identified three subgroups similar to our discovery cohort subgroups I, II and V (Table 2). These subgroups also displayed differences in their 10-year survival outcome from diagnosis (Fig. 5d). Those characterised as subgroup I based on their clinical features (corresponding to the low Noggin and immunoglobulin expression subgroups from RNAseq) (n = 96) demonstrated the lowest survival of 71% from the time of diagnosis. Subgroup V (corresponding to the immune neutral, intermediate RNAseq subgroup) (n = 31) also had an intermediate survival of 86%, while patients in Subgroup II (corresponding to the best surviving subgroup with upregulated Noggin and immunoglobulin genes) showed a very high survival rate of 97.2% (n = 96). These results provide key validation of the existence of endophenotypes for the three major subgroups of patients within the H/IPAH clinical classification group, and that these new subgroups can be identified using routinely collected clinical features associated with RNA dysregulation.

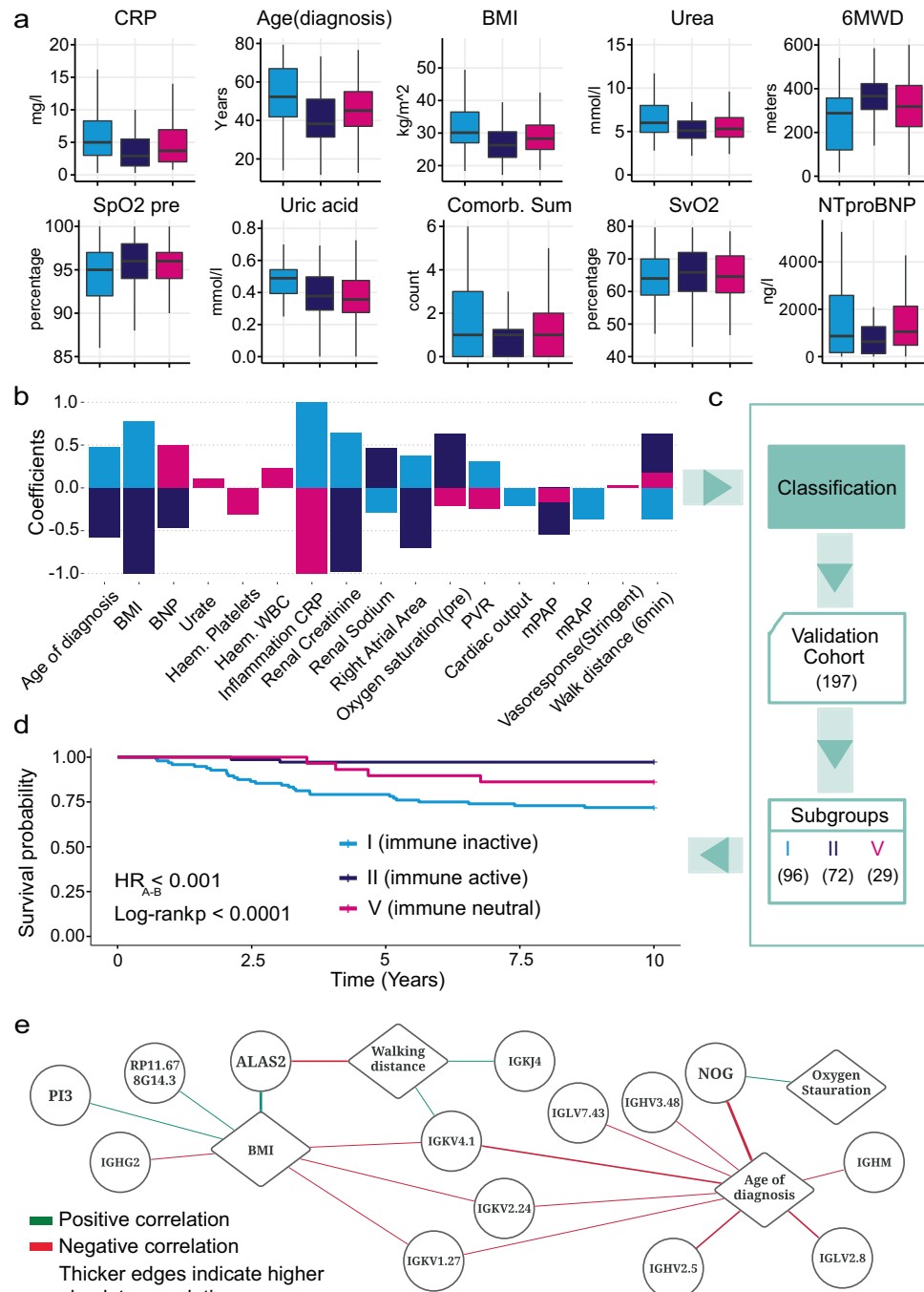

**Fig. 5 Clinical variables descriptive of RNA subgroups and used for classification of new patients. a** Comparison of clinical variables deemed most important from our univariate feature selection model across subgroups I ($n = 129$), II ($n = 112$) and V ($n = 89$). Vertical centre line represents the median, top and bottom bounds of the box represent the first and third quartile, while the tips of the whiskers represent min and max values. **b** Clinical variables selected by ensemble feature selection from models predictive of each subgroup. Coefficients shown for each variable are from the most predictive support vector machine classifiers. **c** Selected clinical features are used to classify 197 IPAH patients from an independent validation cohort. **d** Kaplan–Meier survival curves per predicted subgroup in the validation cohort confirming the difference in survival outcomes between subgroups along with log-rank test $p$ values. **e** Gene and clinical variable correlation network. Diamond nodes represent clinical variables drawn from the clinical signatures. Round nodes represent genes drawn from the gene signature generated by our LASSO model. Edges denoted Spearman rank correlation and have been thresholded to 0.25 and two-tailed test $p$ value $< 1.11 \times 10^{-5}$. Specifically, $corr_{BMI-ALAS2} = 1.27 \times 10^{-11}$, $corr_{BMI-PI3} = 3.17 \times 10^{-6}$, $corr_{BMI-IGHG2} = 4.13 \times 10^{-6}$, $corr_{BMI-RP11.678G14.3} = 8.22 \times 10^{-6}$, $corr_{BMI-IGKV1.27} = 9.32 \times 10^{-6}$, $corr_{BMI-IGKV2.24} = 3.09 \times 10^{-6}$, $corr_{BMI-IGKV4.1} = 9.55 \times 10^{-7}$, $corr_{6MWD-IGKV4.1} = 2.83 \times 10^{-6}$, $corr_{6MWD-IGKJ4} = 2.08 \times 10^{-6}$, $corr_{6MWD-ALAS2} = 7.52 \times 10^{-10}$, $corr_{AoD-IGHV2.5} = 3.72 \times 10^{-10}$, $corr_{AoD-IGLV2.8} = 1.06 \times 10^{-9}$, $corr_{AoD-IGHM} = 6.2 \times 10^{-8}$, $corr_{AoD-NOG} = 3.18 \times 10^{-17}$, $corr_{AoD-IGHV3.48} = 7.7 \times 10^{-7}$, $corr_{AoD-IGLV7.43} = 1.04 \times 10^{-6}$, $corr_{AoD-IGKV4.1} = 6.35 \times 10^{-10}$, $corr_{AoD-IGKV2.24} = 4.19 \times 10^{-6}$, $corr_{AoD-IGKV1.27} = 3.93 \times 10^{-7}$, $corr_{OxygenSat-NOG} = 1.11 \times 10^{-6}$.

**Table 2 Major clinical characteristics of the three subgroups within the validation cohort (n = 197) at time of diagnosis.**

|  | Subgroup I | Subgroup II | Subgroup V | All patients |
|---|---|---|---|---|
| n | 96 | 70 | 31 | 197 |
| Age, years | 65 [55–74] | 40 [29–49] | 45 [30–63] | 54 [39–67] |
| Gender: female | 56 (58%) | 58 (83%) | 18 (58%) | 132 (67%) |
| Vasoresponse | 7 (16.3%) | 8 (21.6%) | 2 (12.5%) | 17 (17.7%) |
| **Treatments** | | | | |
| Phosphodiesterase 5 inhibitors (PDE5i) | 23 (29.5%) | 10 (18.5%) | 8 (29.6%) | 41 (25.8%) |
| Endothelin receptor antagonist (ERA) | 3 (3.85%) | 5 (9.26%) | 1 (3.70%) | 9 (5.66%) |
| PDE5i & ERA combination | 48 (61.5%) | 29 (53.7%) | 17 (63.0%) | 94 (59.1%) |
| Prostacyclin agonist | 1 (1.28%) | 2 (3.70%) | 1 (3.70%) | 4 (2.52%) |
| Calcium channel blockers | 3 (3.85%) | 8 (14.8%) | 0 (0.00%) | 11 (6.92%) |
| **WHO functional class** | | | | |
| I | 4 (4.2%) | 11 (15.7%) | 2 (6.5%) | 17 (8.6%) |
| II | 22 (22.9%) | 26 (37.1%) | 15 (48.4%) | 63 (32.0%) |
| III | 60 (62.5%) | 32 (45.7%) | 12 (38.7%) | 104 (52.8%) |
| IV | 10 (10.4%) | 1 (1.4%) | 2 (6.5%) | 13 (6.6%) |
| 6-minute walking distance, m | 306 (152) | 419 (123) | 409 (120) | 360 (148) |
| NT-proBNP, ng/l | 492 [196; 1327] | 188 [90.0; 400] | 266 [128; 499] | 303 [128; 1092] |
| Forced expiratory volume [% predicted] | 84 (21) | 90 (19) | 90 (17) | 87 (20) |
| Forced vital capacity [% predicted] | 94 (21) | 98 (20) | 99 (15) | 96 (20) |
| Transfer factor of lung for carbon monoxide [% predicted] | 92 (15) | 98 (17) | 96 (11) | 95 (15) |
| **Diagnostic Right Heart Catheter Study** | | | | |
| Mean pulmonary artery pressure, mmHg | 50 [43; 57] | 48 [42; 58] | 49 [41; 57] | 49 [42; 57] |
| Mean right atrial pressure, mmHg | 9 [7; 12] | 7 [5; 10] | 6 [3; 7] | 8 [5; 12] |
| Mean pulmonary Arterial wedge pressure, mmHg | 10 (3) | 8 (4) | 8 (4) | 9 (4) |
| Cardiac Index, l/min/m | 2.0 [1.7; 2.5] | 2.1 [1.7; 2.5] | 2.0 [1.8; 2.4] | 2.0 [1.7; 2.5] |
| Pulmonary vascular resistance, Wood Units | 11 [7; 14] | 11 [9; 15] | 12 [10; 14] | 11 [8; 15] |

Intervals describe first and third quartiles. Parentheses describe standard deviation (SD).

**Clinical signatures are associated with subgroup-specific genes.** We assessed the relationship between gene and clinical features of the subgroups by measuring the correlation between the most predictive features in both signatures. Immunoglobulins IGHV2.5, IGKV4.1, IGLV2.8 and IGHM (Spearman rho = −0.354, -0.342, −0.334, −0.297, respectively, $p$ value <$1.11 \times 10^{-5}$) are negatively correlated with age of diagnosis (Fig. 5e). Indeed, we observed lower expression of immunoglobulin in poor prognosis subgroup I where there were older patients. Noggin was negatively correlated with age of diagnosis (rho = −0.443) but positively correlated with oxygen saturation (rho = 0.275). Interestingly, ALAS2 correlated most strongly with BMI (rho = 0.382) but showed an inverse correlation with 6MWD (rho = −0.323). This is consistent with our observations in the poor prognostic subgroup I where patients with higher expression of *ALAS2* also had higher BMI and shorter walk distances. Genes negatively correlated with BMI included immunoglobulins (*IGKV4.1*, *IGKV2.24* and *IGKV1.27*).

**Gene expression in clinical-feature defined subgroups.** Although the RNAseq whole transcriptome was not measured in this internal validation cohort, we compared gene expression differences between subgroups in this cohort using TaqMan PCR for 17 of the 27 genes (GAPDH used as the endogenous control gene) previously associated with the subgroups and/or clinical variable correlations. Nine of the 11 genes we measured demonstrated a fold change between subgroup I and II in the same direction as the discovery cohort (Fig. 6a). Differences in expression of key genes (*IGHM*, *IGKV2.24*, *IGLV6.57* and *NOG*) were significant ($p < 0.01$) between subgroups I and II (Fig. 6b and Supplementary Table 4).

**External validation of gene and clinical feature correlations.** The correlations between gene and clinical features observed in the discovery cohort were also examined in our validation cohort

of 91 subjects, and also in an external cohort of 32 subjects with RNA collected from PBMCs[26]. We found that 64 of the 90 (71%) correlations measured in these two independent cohorts were consistent with our discovery cohort (Supplementary Table 5).

## Discussion

In this study we describe a machine learning approach to identify transcriptome associated subgroups or endophenotypes of patients with heritable or idiopathic PAH. We defined five distinct clinical subgroups based on clinical presentation, severity and survival. The three largest subgroups displayed significantly different clinical characteristics, severity and survival outcomes suggesting that a molecular classification for PAH may be possible. We also identified patients that progressed through these subgroups over time with treatment and disease progression, the majority of which remaining within their subgroup with only a few transitioning to and from the intermediate subgroup V. The dysregulation of immunoglobulin genes, *NOG* and *ALAS2*, were most predictive of the subgroups with the best and worst prognosis, suggesting that these genes are key in determining patient outcome, and may therefore represent future drug targets but also a tool to identify patients responsive to current treatments. Estimates of cell counts in whole blood revealed elevated levels of lymphocytes, in particular T cells, and lower levels of inflammatory markers in the better prognosis subgroup. We further generated classifiers based on associated clinical features of these new RNA subgroups and used it to identify subgroups that differed in survival outcome in an independent cohort.

The most striking difference between the best and worst surviving subgroups was in immunoglobulin transcription. The upregulation of transcripts coding for the variable domain of immunoglobulin light chains (*IGLV* and *IGKV* genes) that participate in antigen recognition were markers of subgroup II, while their downregulation were markers of subgroup I. Differential levels of *IGVL* and *IGKV* gene transcripts, as seen in subgroup I,

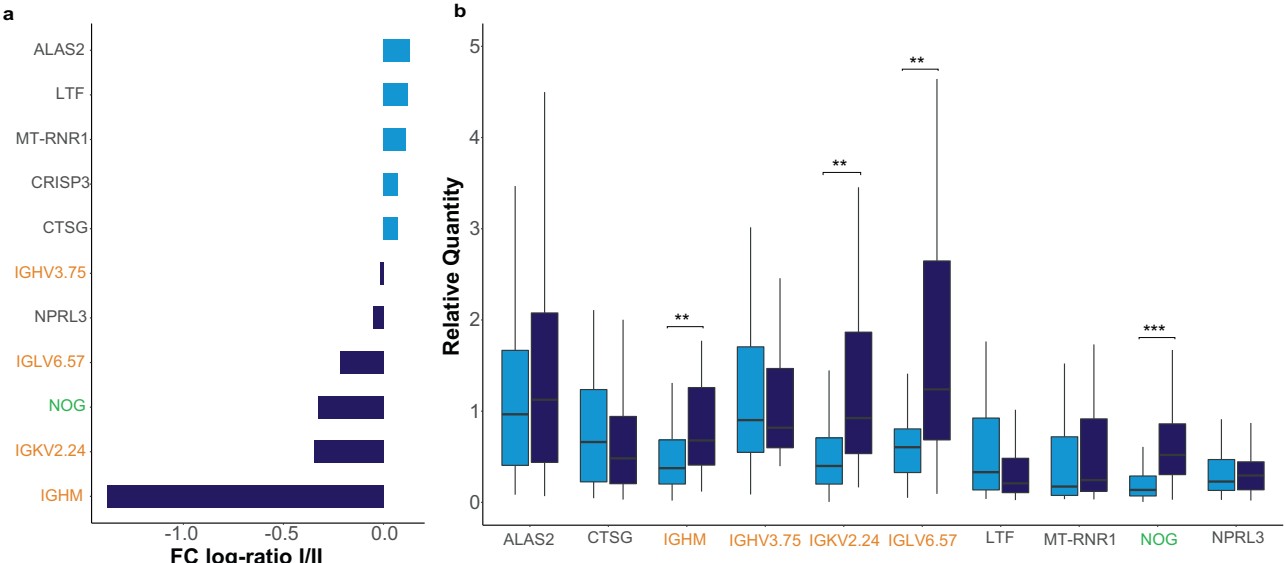

**Fig. 6 Genes of interest with data based on our qPCR results of 91 patients (I = 53, II = 38) of the validation cohort. a** Mean expression fold change (log2 scaled) of the signature genes between validation subgroup I (immune inactive) and II (immune active). The fold ratio was generated based on negative delta Ct values (vs GAPDH). Genes over-expressed in subgroup I are denoted by light blue bars while genes primarily expressed in subgroup II are represented by dark blue bars. **b** The relative quantity (RQ) of each gene of interest relative to GAPDH using a two-sided $t$-test with medians and significant differences shown with $p_{I-II\ (IGHM)} = 8.256 \times 10^{-3}$, $p_{I-II\ (IGKV2.24)} = 2.373 \times 10^{-3}$, $p_{I-II\ (IGLV6.57)} = 5.908 \times 10^{-3}$ and $p_{I-II\ (NOG)} = 1.233733 \times 10^{-4}$. $**p < 0.01$, $***p < 0.001$. Vertical centre line represents the median, top and bottom bounds of the box represent the first and third quartile, while the tips of the whiskers represent min and max values.

may control self-reactivity of human antibodies[27], and the reduction in the diversity of light chains has been associated with several autoimmune diseases, including systemic lupus erythematosis (SLE), type 1 diabetes, and myasthenia gravis[28, 29]. The association between autoimmunity and PAH has long been discussed. There are known associations with autoimmune diseases in other forms of PAH such as systemic sclerosis, SLE, Sjogren's, etc., and the dysregulation of immune cells including T cells, B cells[30] and natural killer cells[31] are well described in IPAH, further validating that our unbiased approach has identified important subgroups. While we detected significant differences in lymphocyte, neutrophil and CRP levels in the blood samples of subgroup I patients, deeper genomic characterisation of T cell receptor and B cell receptors may be needed to understand the role of adaptive immunity on PAH progression.

Beyond the differences in immunoglobulin genes, the expression patterns that defined each subgroup also highlighted haeme biosynthesis through *ALAS2* was a marker for subgroup I and correlated with greater disease severity. Previous gene expression studies across multiple forms of PH, including IPAH, showed significantly increased expression of *ALAS2* in both systemic sclerosis-associated PAH (SSc-PAH) and IPAH[32]. In that study, in IPAH patients increased *ALAS2* levels also demonstrated strong correlation with right atrial pressure, pulmonary vascular resistance, pulmonary artery saturation and cardiac index[32]. These data, and our own observations (Fig. 3), are suggestive of a role for *ALAS2*, iron[33] and hepcidin[34, 35] in pulmonary vascular remodelling and PH. Subgroup II with better prognosis can be partially defined by the downregulation of *ALAS2* and increased expression of *NOG*, a BMP antagonist with high-affinity binding to BMP4 (ref. [36]) which has been shown to inhibit hypoxia-induced proliferation of PASMC[23], and previously associated with BMI in PAH[37] has been proposed as a potential therapeutic target[38]. The role of Noggin in the low-risk group is particularly interesting given the proposed role of both Gremlin and Noggin

in the mechanism of action for Sotatercept in the treatment of PAH[39].

Previous studies have identified clinical features collected during the diagnosis of PAH that also have prognostic utility. The clinical features identified here share many commonalities with those previously included in widely used risk scores (e.g. REVEAL, ERS) assessment for PAH, including, for example, 6-MWD, WHO functional class, and NT-proBNP[4, 40, 41]. This provided further validation that the transcriptomic profile associated with these subgroups provide insight into the biology of disease, and perhaps future drug targets. In addition to biomarkers such as CRP which is known to be elevated in PAH and CTEPH and shown to be predictive of outcome and sensitive to therapies[42] and NT-proBNP with high levels highly prognostic of right ventricular failure[43], age of diagnosis, BMI and renal function were also identified. Renal function has previously been associated with outcome in PAH, although likely because of cardiac function[44]. The age of diagnosis is often discussed as a consequence of genetics[45], or occurrence of co-morbidities; however, in our study the age of diagnosis was most strongly associated with the immunoglobulin light chain genes and Noggin. Carriers of *BMPR2* mutations often present with PAH at a younger age and have a worse survival[46] so the association with Noggin is interesting in the context of perturbed BMP signalling. However, the patients with *BMPR2* mutation did not cluster within one subgroup perhaps fitting with the concept that it is dysfunctional TGFβ/BMP signalling rather than the precise mutation that is important.

There is a well-described sex-paradox in PAH[47] with a 4:1 female to male prevalence but the worse survival in male patients[48, 49]. During our initial analyses of the RNAseq data, we identified subclusters exclusively defined by sex genes. To mitigate against any gender bias, we excluded sex-chromosome-associated genes in our preprocessing steps of the analysis pipeline. Although we cannot reject the possibilities of the

aforementioned genes contributing towards PAH or resilience, we believe that their removal ensures that the clustering algorithm captures heterogeneity independent of sex-associated expression variation. However, the interactions between gender and other autosomal genes in the context of PAH require further study.

The application of unsupervised learning from molecular profiles of IPAH is a powerful approach for revealing subgroups within a heterogeneous population that has not been defined clinically. Most studies employ widely used clustering algorithms without exploring their data suitability. By contrast, in this study we determine spectral clustering as the most consistent method in detecting differences and subsequently partitioning RNA-sequencing samples using robust performance criteria. Furthermore, previous studies have focused on clustering all PAH cases using a small set of immune markers, and captured immune phenotypes overlooked by the broad clinical classifications[50]. We used a much larger set of features, i.e., the whole transcriptome and clustered cases lacking causal pathologies, and also found immune phenotypes that differentiated the subgroups. While we controlled for confounding factors that affect clustering, such as gender-associated genes (Supplementary Fig. 1), there may yet be other hidden factors, such as viral infections related to age and gender that could influence patterns observed from whole blood[51]. The large degree of validation of the subgroups using both transcriptomic and clinical features to define them provides strong evidence that these endophenotypes are reproducible and may be useful to risk stratify or biologically classify subgroups of IPAH patients. However, further transcriptomic studies profiling patients at multiple timepoints are required to fully understand the dynamics of the immune components we identified, the frequency of acute infections, and the impact on PAH phenotype.

Transcriptomic profiling of the blood samples coupled with clinical data from IPAH patients provides an insight into endophenotypes that may describe this heterogeneous disease based on RNA expression. The use of additional 'omic' biomarkers to provide further molecular profiles (e.g. DNA, protein, metabolites) as stable biomarkers for stratifying patients could further improve our algorithmic predictions of patient outcomes and reveal endophenotypes to be targeted therapeutically. Furthermore, these data hold promise that these molecular endophenotypes may be tractable to existing therapies, may offer an alternative approach to tailor, and assess individual treatment response, in PAH as well as offering insights into disease pathogenesis that can be targeted by therapeutics as a precision medicine approach[52] in PAH and potentially other diseases to drive molecular clinical classification suited to the future precision medicine era in healthcare.

## Methods

**Study design.** The Cohort study of idiopathic and heritable PAH is an observational, prospective and longitudinal study of patients with idiopathic and heritable PAH (clinicaltrials.gov NCT019072950). The Sheffield Teaching Hospitals Observational Study of Pulmonary Hypertension, Cardiovascular and other Respiratory Disease (UK REC Ref 18/YH/0441) is a longitudinal study of patients with suspected pulmonary hypertension or an associated cardiovascular or respiratory condition. Follow-up information is collected as a part of routine clinical care every 6 months. The study allows recruitment of both incident and prevalent cases. Patients consented to the study agreed to have blood taken for next-generation sequencing and other omics studies. Healthy adult controls were recruited for comparison studies. The subsequent whole-blood sample collection process is described in ref. [13].

**Ethics.** All UK samples were obtained following informed consent into the UK National Cohort Study of Idiopathic and Heritable Pulmonary Arterial Hypertension (clinicaltrials.gov NCT01907295; UK REC Ref. 13/EE/0203) and/or the Sheffield Teaching Hospitals Observational Study of Pulmonary Hypertension, Cardiovascular and other Respiratory Disease (UK REC Ref 18/YH/0441). Data were obtained from samples collected at the University of Arizona Pulmonary Hypertension clinic between 2012 and 2015 following institutional guidelines and following informed consent.

**Participants.** Patients diagnosed with I/HPAH, PVOD or PCH, relatives of index cases and unrelated healthy controls were recruited at nine UK centres and followed up by a median of 7.9 years. In total, 358 patients (Supplementary Fig. 11) of which 96.7% were further verified to be I/HPAH, 13 relatives, and 21 healthy controls recruited to the I/HPAH Cohort study were analysed. Both prevalent and incident cases were allowed. Prevalent cases were defined as diagnosed earlier than 6 months before the study initiation. Patients in Cohort study were followed longitudinally as part of their clinical PAH care. All cases were diagnosed between March 1994 and November 2016, and diagnostic classification was made according to international guidelines[53]. Patients with PAH associated with anorexigen exposure were considered as IPAH, whereas HPAH was defined by the presence of a positive family history of PAH. Clinical, functional and haemodynamic characteristics at the time of PAH diagnosis were prospectively entered into the database. The date of diagnosis corresponded to that of confirmatory right heart catheterisation.

Following diagnosis, subsequent treatments and follow-ups were at the discretion of the treating physician, according to the contemporary guidelines. In most centres, patients were seen every 3–6 months with an assessment of functional status and exercise capacity. Right heart catheterisation was repeated when considered necessary by the responsible clinician. Study visits were performed every 6 months. Healthy controls had been sampled only once and had clinical information recorded from the time of sampling.

**Clinical data capture, processing and quality control.** Pseudonymised results of routinely performed clinical tests reported in either clinical case notes or electronic medical records (EMR) were stored in web-based OpenClinica (OC) data capture system (Community edition). Twenty electronic Clinical Case Report Forms (eCRFs) distributed across seven events (Diagnostic, Continuous data, Follow-up, Epidemiology questionnaire, Suspension, Relatives, Unrelated healthy control) were constructed to accommodate routinely available clinical information. Details regarding data verification procedures were previously described in detail[54].

Information about participants' status was collected every 6 months (via National Health System Digital Spine portal or an equivalent local system). Current analysis was performed on the census performed on 31 January 2020. Two risk assessment strategies were applied to the data. Reveal risk score[4] and abbreviated ERS risk scores[55] were calculated in all patients who had the necessary minimum phenotypic information available. Patients who died or were transplanted were suspended on the day of the event, patients who withdrew from the study were censored on the date of the last visit, the reason for withdrawal was recorded.

**Missingness assessment and imputation.** Missingness rates, patterns and causes were assessed per individual, variable and centre and visualised with vim package v5.1.1R (Supplementary Fig. 12). Multiple imputation by the chain equations method was used to impute missing data (mice v3.8.0 package R)[56]. The imputation model included all variables that were necessary in the analysis model, including cumulative baseline hazard function and variables that predicted both the incomplete variable and if the incomplete variable was missing like the centre and whether the case was incident or prevalent. Quality of predictors was assessed using outflux–influx plot. Numerical data were imputed with predictive mean matching (pmm), factors with two levels were imputed using logistic regression, factors with more than two levels with multinomial logit model and ordered factors with more than two levels with the ordered logit model. Transformed variables (BMI, ratios, score sums) were imputed as just another variable as well as passively with good concordance. The visiting sequence was set to 'monotone' to speed up convergence. The number of iterations was set to 20. Following the rule of thumb proposed by White et al.[57] that the number of imputations should be at least equal to the percentage of incomplete cases, the procedure was performed at $m = 50$. The convergence of the algorithm was checked, and the means and standard deviations of imputed values were plotted over 20 iterations. The streams of numerical and factor variables intermingled and showed no trends at later iterations. Factors influencing the accuracy of the imputation include the variability in time between diagnosis and sampling, higher missingness in clinical data for prevalent cases (diagnosed sometimes many years ago), and differences in measurement error between centres which followed different protocols for clinical data collection.

**RNA data preprocessing.** A number of preprocessing steps were required to prepare the raw sequencing data for unsupervised machine learning. High-throughput sequencing generated raw pair-end counts of 205,259 transcripts across 508 samples that belong to GenCode Release 28 (GRCh38.p12). Consequently, Salmon (https://combine-lab.github.io/salmon/) was used to estimate the relative abundance of the transcripts (TPM, units of transcripts per million) which were then mapped to genes ($n = 60,144$) using the tximport R package. Only genes with more than two reads (in a transcript level) in at least 95% of control and patient samples were considered and 11 additional male genes were removed ($n = 25,955$). Hyperbolic arcsine transformation (package base v3.6.0) was applied to the final RNAseq TPM matrix. Further information on quality control of samples and genes

can be found in the Supplementary Methods. The RNA-sequencing and clinical data of healthy controls were not used in the main pipeline of this study. A secondary clustering with all patient and healthy samples was implemented to demonstrate the lack of pure patient and healthy subgroups within our cohort (Supplementary Fig. 6b). Principal component analysis of expression profiles from samples with a second replicate clustered together according to the first four principal components (Supplementary Fig. 13).

**Spectral clustering: gene expression subgroup identification**. We performed cluster analysis to partition IPAH patients to distinct RNA-based groups. The spectral clustering model (package kernlab v0.9-29) was selected as the most suitable unsupervised learning algorithm based on the highest partitional consistency when comparing multiple dissimilar algorithms (Supplementary Table 6). For the spectral clustering method, data points (i.e. patients) are embedded and partitioned in a low-dimensional space in the form of a similarity graph, rather than being characterised by more than 25,000 gene dimensions. High partitional consistency was defined as the high adjusted Rand Index (package fossil v0.3.7) and low standard deviation calculated between different variations of each clustering algorithm (k-means, spectral, hierarchical clustering), as described in Clustering algorithm selection. For the selection of the most appropriate clustering algorithm we utilised 25,955 genes across 359 IPAH patient samples (discovery cohort) after further filtering for repeated same-visit samples and non-H/IPAH diagnosis. We compared three fundamentally different methods (hierarchical, k-means and spectral) and use partitioning consistency to determine which method picks up an underlying signal from our data type (RNA-sequencing). As shown in Supplementary Fig. 14, spectral clustering showed the highest consistency (Adjusted Rand Index) in detecting differences and subsequently partitioning patients in similar clusters independently of the kernel. Notable is the difference in intra-agreement of spectral (~75%) and k-means (−13%) clustering, which highlights the importance of the extra step of mapping data in a low-dimensional space (as a similarity graph) in spectral clustering. To run the main spectral clustering partitioning we first selected the most relevant gene set by ranking all genes based on the variability of their expression across patient samples using the stats v3.6.0R package (Supplementary section 'Feature selection of genes'). Subsequently, several candidate gene sets of increasing size were drawn from the top ranking gene list and the one that generated subgroups of highest stability, according to package fpc v2.2-3, was selected (Supplementary section 'Highest stability gene set'; Supplementary Fig. 15). This resampling bootstrap approach determined that the most stable gene set was composed from the 300 most variable genes. For the secondary clustering run with only IPAH, 1700 variable genes were selected as the most stable gene set for clustering.

The number of IPAH subgroups was estimated through ensemble learning[58] utilising 15 internal indexes calculated using the package diceR v0.6.0 (Supplementary sections 'Optimal number of subgroups k' and 'Internal Index Voting'). A representation of patient flow across k can be found at Supplementary Fig. 16. The Radial Basis function kernel was used as the similarity measure with five target subgroups, identified as the optimal number of subgroups by an ensemble learning method. We elected to investigate k = 5 subgroups, since in clustering contexts it is safer to overestimate than underestimate the number of subgroups to prevent loss of information. However, k = 3 subgroups were voted from the vast majority of methods and we expect them to be the main subgroups. Further information on the selection of clustering algorithms and parameters can be found in the Supplementary Methods.

**Analysis of subgroup differences**. Survival analysis was performed (R package survival 3.1-7) on the main (Supplementary Fig. 17) and validation cohorts to identify the survival differences between subgroups. Kaplan–Meier survival curves from diagnosis and sampling were calculated for the main patient cohort (per spectral subgroup) as well as the validation cohort (per predicted subgroup). Subsequently, two multivariate Cox models were fitted and Hazard ratios calculated on the main cohort once adjusting for gender and once adjusting for the composite clinical signature discovered by supervised machine learning. Gene signatures for each subgroup were identified using LASSO regression models with cross-validation (package glmnet 3.0-1). The variables with the 5% highest coefficients for each class were highlighted (Supplementary Fig. 18), and the full list of non-zero coefficients for each class can be found in Supplementary Data 3. The pathfinder R package was used to highlight enriched gene pathways between subgroups and differential expression analysis using the DESeq2 package[59] was performed on genes associated with the subgroups (Supplementary Fig. 19). Gene expression differences across subgroups are presented in Supplementary Fig. 20. All statistical tests between subgroups were two-sided and Bonferonni adjusted for multiple testing.

**Identifying clinical signatures of subgroups**. The dataset was initially cleaned and filtered on 119 features that were identified by a domain expert from the original 887 features that described the dataset. Subsequently, any feature that had more than 5% missing data was dropped, and categorical features numerically encoded.

All ML tasks were carried out using Scikit-learn[60] ML framework version 0.23.2 in a Python 3 environment. As machine learning classifiers, we used Logistic Regression (LR), support vector machines (SVM), Random Forest (RF) and k-nearest neighbour (kNN). RF is a powerful ensemble learning technique especially for high-dimensional classification tasks. Further details about classifier training and feature selection can be found in the Supplementary Methods.

**Classification of new patients using signatures**. Each clinical signature was used to develop a classification model trained on the discovery cohort to classify new patients into the RNA-based subgroups. Classification models were built using SVM[61], RF[62], LR[63] and KNN[64]. The candidate signature that obtained the best performance was selected. This process was repeated for all signature sizes, s = 1 to s = 20, for subgroups I, II and V. A final signature for each subgroup was selected based on a compromise between the fewest number of features (s = 1 to s = 20) and classification performance. Final selected signatures for each of the subgroups were pooled to create a composite signature, which was then used in a multi-class classification model. The model was trained on the discovery dataset to discriminate between subgroups I, II and V, used to predict subgroup membership of an unseen validation dataset. The predicted subgroup membership was then used to calculate survival of predicted subgroups. Survival of the predicted subgroups was compared to known survival of subgroups in the discovery dataset for validation purposes.

**qPCR on validation cohort**. Frozen Tempus tubes collected from patients in the validation cohort, collected under the UK National Cohort study, were obtained; RNA was extracted using Maxwell® 16 LEV simplyRNA Blood Kit (Cat.# AS1310) as described in the manufacturer's instructions on the Maxwell® 16 Instrument (Cat.# AS2000). Extracted RNA was transcribed using the High-Capacity-RNA-to-cDNA kit (Thermo Fisher Cat.# 437406) following the manufacturer's instructions. Resultant cDNA was analysed using custom TaqMan array cards (Thermo Fisher Cat.#4342249) with Fast Advanced Mastermix (Thermo Fisher Cat.# 4444964); damples were run 8 to a card across 25 cards with 24 primer probes (Thermo Fisher) per sample (18S-Hs99999901_s1, ACTB-Hs00357333_g1, ALAS2-Hs01085701_m1, BMPR2-Hs00176148_m1, C4BPA-Hs00426339_m1, CRISP3-Hs00195988_m1, CTSG-Hs00175195_m1, GAPDH-Hs02786624_g1, HPRT1-Hs02800695_m1, IFI27-Hs01086373_g1, IGHM-Hs00941538_g1, IGHV3-75-Hs03832008_sH, IGKV2-24-Hs06671746_g1, IGLV6-57-Hs01696637_s1, LINC00221-Hs01382601_m1, LTF-Hs00914334_m1, MT-RNR1-Hs02596859_g1, NEBL-Hs01067284_m1, NOG-Hs00271352_s1, NOS2-Hs01075529_m1, NPRL3-Hs00429221_m1, PI3-Hs00160066_m1, SMIM11A;SMIM11B-Hs00938773_m1, XIST-Hs01079824_m1). These assays were performed in duplicate using the Applied Biosystems 7900HT Fast real-time PCR system with the TaqMan Low Density Array card block following calibration using the TaqMan Low Density Array Calibration Kit (Thermo Fisher Cat.# 10341465). Ct values were determined with Automatic thresholding in the SDS2.4 software. GAPDH- Hs02786624_g1 was used as a control. Relative quantity was calculated using the ΔΔCt method.

**External cohort validation**. An external validation cohort of patients with Group 1 PAH prospectively recruited at the University of Arizona Pulmonary Hypertension clinic between 2012 and 2015 following institutional guidelines and informed consent was used. The cohort comprised 84 subjects with Group 1 PAH of whom 32 were diagnosed with idiopathic PAH. For each subject, demographics and clinical variables were collected[26]. PBMCs were stored in RNAlater as previously described.[8] In total, approximately 3600 million clusters with paired-end 75 bp reads (~35M cluster per sample) were generated from PBMC-derived RNA.

**Clinical variable and gene correlations**. We calculated correlations between the clinical and gene signatures we generated in previous steps of this study. For discovery and validation cohorts we used the rcorr function of R package Hmisc (version 4.5-0). For the external validation we used the values found in ref. [26].

**Study approval**. Study approval for the use of sample and data were obtained from the UK National PAH Cohort Study Data Access Committee (clinicaltrials.gov NCT01907295; UK REC Ref 13/EE/0203), and the Sheffield Teaching Hospitals Observational Study of Pulmonary Hypertension, Cardiovascular and other Respiratory Diseases Scientific Advisory Board (UK REC Ref 18/YH/0441).

**Reporting summary**. Further information on research design is available in the Nature Research Reporting Summary linked to this article.

# Data availability

The transcriptomic and clinical data used in this study have been deposited in the EGA (the European Genome-phenome Archive) database under accession code EGAS00001005532[65]. In compliance with the Ethics under which these data and samples have been collected, the transcriptomic data are available through restricted access for approved researchers who agree to the conditions of use, i.e. keeping it secure and only using it for approved purposes. To apply for access please contact

cohortcoordination@medschl.cam.ac.uk. You will receive an application form within 30 days. The 'UK National PAH Cohort Study Data Access Committee' will review requests within 3 months of receipt of the completed application form and if approved, provide details for access to the RNAseq data stored at the EGA. All requesters must agree to the data access conditions found in EGA. The data used to generate statistics, plots and figures are accessible through our interactive portal found in https://sheffield-university.shinyapps.io/ipah-rnaseq-app/. Source data are provided with this paper.

## Code availability

Additionally, the code used to generate the results of this study is publicly available at https://zenodo.org/badge/latestdoi/299615578 (ref. [66]).

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

## Acknowledgements

The UK National Cohort of Idiopathic and Heritable PAH is supported by grants from the British Heart Foundation (SP/12/12/29836 & SP/18/10/33975) and the UK Medical Research Council (MR/K020919/1). Additional samples from the Sheffield Teaching Hospitals Observational Study of Pulmonary Hypertension, Cardiovascular and other Respiratory Diseases were supported by British Heart Foundation (PG/11/116/29288). We gratefully acknowledge financial support from the UK Department of Health via the NIHR comprehensive Biomedical Research Centre award to Imperial College Healthcare NHS Trust, Cambridge Biomedical Research Centre, and Guy's and St Thomas' NHS Foundation Trust in partnership with King's College London and King's College Hospital NHS Foundation Trust and the NIHR Imperial Clinical Research Facility. Sheffield NIHR Clinical Research Facility award to Sheffield Teaching Hospitals Foundation NHS Trust. S.K. is supported by a Donald Heath Ph.D. Studentship award; C.J.R. is supported by a British Heart Foundation Intermediate Basic Science Research fellowship (FS/15/59/31839). N.E. is supported by an EPSRC Centre for Doctoral Training; A.A.R.T. is supported by a British Heart Foundation Intermediate Clinical Research fellowship (FS/18/13/33281); N.W.M. is a British Heart Foundation Professor and NIHR Senior Investigator. A.L. is supported by a BHF Senior Basic Science Research fellowship (FS/18/52/33808). E.J. is supported by the Academy of Medical Sciences Springboard (ref: SBF004/1052). M.R.W. is in receipt of a British Heart Foundation Centre for Research Excellence award (RE/18/4/34215). M.J.D. and D.W. are supported by the NIHR Sheffield Biomedical Research Centre. We thank and thank all the patients and their families who contributed to this research, the UK Pulmonary Hypertension Association for their support, NIHR BioResource volunteers for their participation, and gratefully acknowledge NIHR BioResource centres, NHS Trusts and staff for their contribution. We thank the National Institute for Health Research, NHS Blood and Transplant, and Health Data Research UK as part of the Digital Innovation Hub Programme. The views expressed are those of the author(s) and not necessarily those of the NHS, the NIHR or the Department of Health and Social Care.

## Author contributions

SK, EJ and EMS contributed equally. All authors made substantial contributions to the conception or design and data acquisition of the work. S.K., E.J., E.M.S., J.A.P., C.J.R., P.O., J.W., J.I., M.J.D., D.P., T.S.M., N.E., A.A.R.T., C.E.R., F.R., J.G.N.G., J.X.-J.Y., T.-H.S., A.A.D., G.C., J.L., P.A.C., L.S.H., R.C., D.G.K., C.C., J.P.-Z., M.T., S.W., S.G., N.W.M., M.R.W., A.L. and D.W. performed the analysis and/or interpretation of data. S.K., E.J., E.M.S., A.L. and D.W. drafted the work and all authors revised it critically for important intellectual content; and gave final approval of the version submitted for publication; and agree to be accountable for all aspects of the work in ensuring that questions related to the accuracy or integrity of any part of the work are appropriately investigated and resolved.

## Competing interests

The authors declare no competing interests.

## Additional information

## UK National PAH Cohort Study Consortium

Marta Bleda[3], Charaka Hadinnapola[3], Matthias Haimel[3], Kate Auckland[3], Tobias Tilly[3], Jennifer M. Martin[3], Katherine Yates[3], Carmen M. Treacy[3], Margaret Day[16], Alan Greenhalgh[16], Debbie Shipley[16], Andrew J. Peacock[17], Val Irvine[17], Fiona Kennedy[17], Shahin Moledina[18], Lynsay MacDonald[18], Eleni Tamvaki[18],

Anabelle Barnes[18], Victoria Cookson[18], Latifa Chentouf[18], Souad Ali[19], Shokri Othman[19], Lavanya Ranganathan[19], J. Simon R. Gibbs[4,19], Rosa DaCosta[20], Joy Pinguel[20], Natalie Dormand[20], Alice Parker[20], Della Stokes[20], Dipa Ghedia[21], Yvonne Tan[21], Tanaka Ngcozana[21], Ivy Wanjiku[21], Gary Polwarth[13], Rob V. Mackenzie Ross[22], Jay Suntharalingam[22], Mark Grover[22], Ali Kirby[22], Ali Grove[22], Katie White[22], Annette Seatter[22], Amanda Creaser-Myers[23], Sara Walker[23], Stephen Roney[23], Charles A. Elliot[5], Athanasios Charalampopoulos[5], Ian Sabroe[5], Abdul Hameed[5], Iain Armstrong[5], Neil Hamilton[5], Alex M. K. Rothman[2,5], Andrew J. Swift[2,5], James M. Wild[2], Florent Soubrier[24], Mélanie Eyries[24], Marc Humbert[25], David Montani[25], Barbara Girerd[25], Laura Scelsi[26], Stefano Ghio[26], Henning Gall[27], Ardi Ghofrani[27], Harm J. Bogaard[28], Anton Vonk Noordegraaf[28], Arjan C. Houweling[28], Anna Huis in't Veld[28], Gwen Schotte[28] & Richard C. Trembath[29]

[16]Freeman Hospital, Newcastle, UK. [17]Golden Jubilee National Hospital, Glasgow, UK. [18]Great Ormond Street Hospital, London, UK. [19]Hammersmith Hospital, London, UK. [20]Royal Brompton Hospital, London, UK. [21]Royal Free Hospital, London, UK. [22]Royal United Bath Hospitals, Bath, UK. [23]Sheffield NIHR Clinical Research Facility, Royal Hallamshire Hospital, Sheffield, UK. [24]Département de génétique, hôpital Pitié-Salpêtrière, Assistance Publique-Hôpitaux de Paris, and UMR_S 1166-ICAN, INSERM, UPMC Sorbonne Universités, Paris, France. [25]Université Paris-Sud, Faculté de Médecine, Université Paris-Saclay, AP-HP, Centre de référence de l'hypertension pulmonaire sévère, INSERM UMR_S 999, Hôpital Bicêtre, Le Kremlin-Bicêtre, France. [26]San Matteo, Pavia, Italy. [27]University of Giessen, Giessen, Germany. [28]VU University Medical Center, Amsterdam, The Netherlands. [29]Health and Life Sciences, King's College, London, UK.

