## [Peer Review File · Nature Communications]

REVIEWER COMMENTS

Reviewer #1 (Remarks to the Author):

The work presented in this manuscript aimed at identifying idiopathic pulmonary arterial hypertension blood transcriptome endotypes. The disease is rare but often fatal and clinical course can vary widely in this patient population. Blood transcriptomics seems well-suited to the identification of signatures that may be predictive of clinical severity and outcomes.

A strength of the work is obviously the remarkable number of cases with 5-year outcome information available for this study and profiled by this team (all the more remarkable, that this again is a rare disease). Potential clinical/translational impact is therefore possible from such a large cohort and gives this work particular importance.

The authors employed an unsupervised approach to tackle inter-individual variations and identify distinct endophenotypes based on differences in blood transcript abundance measured on a genome-wide scale. Three distinct blood transcriptome patterns were identified, each associated with distinct clinical outcomes (good, poor, intermediate). Clinical features associated with each blood transcriptome endophenotype were then identified and in turn used to define clinical endophenotypes. The corresponding clinical parameters (rather than transcriptomic ones) were then used to classify an independent validation set and confirm their predictive value.

Here are some comments:

- 1) The point of using blood transcriptomics seems largely lost in this particular workflow. Would it not have been better indicated, in order to identify clinical parameters predictive of outcome (the endpoint of this analysis), to simply use such clinical parameters in the training/discovery set? A higher level of performance would presumably thus be achieved. As it is, the blood transcriptome data is solely used for the purpose of identifying signatures associated with outcomes (pathogenesis), rather than identifying predictive biomarkers. This altogether does not appear very logical. But short of completely changing their approach the authors should at least clarify this point. Or they may consider omitting the last part of the analysis (validation of the predictive performance of clinical classifiers), which tends to undermine the premise of the work.
- 2) It is not apparent from reading the manuscript at first, but the authors are actually re-analyzing a dataset that they had previously generated and published results from using a different analytical workflow [ref 13]. Data reuse should definitely be encouraged, and it is not per se a negative point

here. But some good practices would also need to be adopted. One would be to be upfront about it and clearly state that the data is being reused. A reason for that is that there also needs to be a discussion of the findings that have been reported earlier, and of the motivation of the re-analysis. What analysis gaps are being addressed? How do these findings and conclusions differ from those that were drawn earlier?

3) What compounds point #2 is that the authors had not made their data publicly available upon publication of their earlier work [ref 13], as is the custom and indeed a mandate in most quality journals. And they pass up the opportunity here once again, stating that data “are available from the corresponding author upon reasonable request”. Which practically is far from guaranteeing that the data will indeed be shared with the community for further re-use. Generating these data is obviously the culmination of many years of work and it is understandable that the authors wish to capitalize on those as much as possible. But after the publication of a secondary analysis, it seems that it would be indicated to make these data public and potentially further extending the impact of their work (and benefit to patients with this disease).

4) An earlier study by a different group also associated the severity of IAH with high levels of expression of ALAS2 [ref 31]. This work would deserve to be discussed in more detail here (only one sentence as it is).

5) Addressing inter-individual variation employing unsupervised clustering approaches is suitable, although the approach has been employed for decades. Some novelty can be claimed concerning implementation itself, but dozens of clustering methods could be used with relatively similar outcomes. This is not a major criticism; it might just be indicated to downplay the novelty of the analytical approach accordingly.

Reviewer #2 (Remarks to the Author):

The authors propose a machine-learning-based analysis of gene expression data for idiopathic pulmonary arterial hypertension classification. The authors’ goal is to use a combination of unsupervised and supervised learning to identify subtypes of patients with different risk profiles and so survival rates. Identification of the appropriate patient subgroups is an important problem. A novel, robust, and accurate method will be invaluable. Thus this manuscript addresses a critical challenge. The concept of this work is interesting, but there are a number of issues that need clarification.

1. How exactly an independent cohort was defined? Was it profiled in a different lab? Gene expression measurements are prone to batch-effect issues. I suggest the authors explore the performance of the selected subset of features/ signatures (clinical parameters+genes) and the model built on it on the completely independent dataset. GEO or ArrayExpress can be used for such purposes.

2. HR and Cox regression were not adjusted to age, but the only gender. While it is seen that low-risk groups are somehow younger intermediate and high-risk groups and risk is growing with age. What was the age of healthy controls? Age is one of the important factors for survival in general. Could it be the case for the better survival of group II vs groups I and V? I suggest reporting sex and age-adjusted HR.

3. Related to the previous point, it would be interesting to see how the standard clinical features would perform in risk prediction in comparison to gene expression profiling. Would it be possible to get similar clusters of patients with K-means/ Hierarchical clustering/Spectral clustering for their clinical features only?

4. Names of packages used (for SVM, RF, and other methods and algorithms used) are lacking

5. Samples are not defined - p 458. It is also confusing what is the difference between features and signatures.

How many gene expression replicas were available per one patient?

6. It is not clear how authors moved from 25,955 genes to 1 to 20 features? What was the final set of genes used to trained supervised models?

7. Figure 3 B: p-values are lacking for selected genes. Were observed differences significant? What was the cut-off for FC?

8. Authors report that some of the features were imputed, but results are not reported. How many of the patients' profiles had missing values?

Reviewer #3 (Remarks to the Author):

The investigators use clustering analyses and machine learning to examine biological and clinical heterogeneity in a cohort of patients with pulmonary arterial hypertension (PAH). They define 5 clusters on the basis of RNASeq data and move forward with describing 3 of the 5 clusters as the other two contain only 29 patients. They find that the 3 main clusters are associated with immunoglobulin levels and different gene expression patterns. The clusters are also associated with differences in their clinical risk as well as long-term outcomes. Focusing on clinical risk factors only,

the clinical factors associated with each of the biological cohorts is validated in a second patient population. This is an interesting study of a relatively large PAH cohort.

1. More information is needed in this paper to understand the study cohort. Were hereditary patients genotyped? Were data from those patients who had a final diagnosis that wasn't PAH (supplementary figure 5) but did have RNASeq done taken out of all of the analyses?
2. Taking out the transcripts that segregated patients by male sex is understandable based on data presented in the supplement, but since you are looking at RNA transcripts at a single point in time, it doesn't discount the possibilities that these are contributors to disease or resilience. This should be mentioned.
3. The paper focuses on 3 of the 5 cluster subgroups since 2 of the clusters contained only 29 patients. This raises the question about whether or not 5 is the optimal cluster number and whether or not it should have been 3. The various methods and measures of distance are shown in the supplement, but not all of these methods are not always appropriate for this type of data. By using each of these methods, did $k=5$ for all or is your internal indexes component of Fig. 1 meant to indicate that you took an average of the results of the other methods? If you perform clustering with $k=3$, what happens to the patients in subgroups 3 and 4 and does it change the transcripts associated with each subgroup? Since you did define 5 clusters, you should report on all of them throughout the Results section.
4. Use of the REVEAL 2.0 risk score is a nice way to describe the cohort. Does adding transcript profiles to this improve the predictive value (as the team has done in other publications)?
5. The inclusion of hereditary and idiopathic PAH in the cohort is interesting, but begs the question about whether they should be analyzed separately. If you look at the idiopathic group alone, do you get the same results?
6. The presence of the immunoglobulin chains is interesting and in line with prior reports about immune cell activation. Given that there are differences between immune and inflammatory cell numbers between the groups, if Ig transcripts are corrected for cell numbers, is there really still a difference between the groups to show actual differences?
7. On page 9, the statement that the average time between sampling and diagnosis was 5.3 years with a range that extends from 0.002-21.57 years is confusing. The way that this is written suggests that the samples were obtained first and the diagnosis of PAH was made later. Yet the sentence before it states that patients were diagnosed at a median of 45 years and sampled at a median of 52 years. Are you trying to indicate disease duration? If so, state accordingly. Also, the duration of time between diagnosis and sampling is vast, so please provide some more granular information on time from diagnosis to sampling. PAH patients with at least 22 years survival are quite different than patients with incident disease.
8. Please also include age at diagnosis in Table 1 for each of the subgroups. Also, major characteristics should also include right heart cath data, RV function, and testing to rule out PE. PFT data alone are isolated descriptors. It also appears that no statistical testing took place.

9. More information is needed to understand how the variables were chosen for supervised machine learning. Was this by the investigators? Based on prior publications? What was significant in the dataset?

10. CRP shows up in all your groups and BMI in several. You don't indicate if the levels are the same for the groups, high vs. low, etc. The levels aren't reported. It's well recognized that there are age, race/ethnicity, and sex differences in values of the factors. How did you adjust for this?

11. How many patients were on disease-modifying drugs, steroids, or anti-inflammatory agents at the time of sampling? Since this is a PAH cohort, it is expected that the CTD patients at a minimum would fall into this category.

12. The clinical features list should be put in table format to make it easier to understand the differences between the groups.

13. The validation cohort study is perplexing since you use RNASeq to identify clinical variables to predict outcomes and then validate the clinical variables. One could make the argument that you could just use the REVEAL risk score to predict outcomes. Is the discriminatory power of the variables identified by your methodology better than the REVEAL risk score in the validation cohort? If the idea is also to identify targets for therapies (last sentence of your abstract), then how does validating clinical characteristics in a separate cohort help this cause? It seems that you would have to show that the validation cohort has relatively the same RNASeq profile.

Thank you for re-reviewing our manuscript and providing thoughtful comments and suggestions. Below is a point-by-point response to the comments/queries raised with the reviewers' comments in italics, our response in plain text and changes in the revised manuscript text in red below, and in the revised manuscript.

Reviewer #1

The work presented in this manuscript aimed at identifying idiopathic pulmonary arterial hypertension blood transcriptome endotypes. The disease is rare but often fatal and clinical course can vary widely in this patient population. Blood transcriptomics seems well-suited to the identification of signatures that may be predictive of clinical severity and outcomes.

A strength of the work is obviously the remarkable number of cases with 5-year outcome information available for this study and profiled by this team (all the more remarkable, that this again is a rare disease). Potential clinical/translational impact is therefore possible from such a large cohort and gives this work particular importance.

The authors employed an unsupervised approach to tackle inter-individual variations and identify distinct endophenotypes based on differences in blood transcript abundance measured on a genome-wide scale. Three distinct blood transcriptome patterns were identified, each associated with distinct clinical outcomes (good, poor, intermediate). Clinical features associated with each blood transcriptome endophenotype were then identified and in turn used to define clinical endophenotypes. The corresponding clinical parameters (rather than transcriptomic ones) were then used to classify an independent validation set and confirm their predictive value.

We thank the reviewer for recognizing the challenges in working with rare diseases, recognising the need to better stratify patients and our approach.

Comments

1. The point of using blood transcriptomics seems largely lost in this particular workflow. Would it not have been better indicated, in order to identify clinical parameters predictive of outcome (the endpoint of this analysis), to simply use such clinical parameters in the training/discovery set? A higher level of performance would presumably thus be achieved. As it is, the blood transcriptome data is solely used for the purpose of identifying signatures associated with outcomes (pathogenesis), rather than identifying predictive biomarkers. This altogether does not appear very logical. But short of completely changing their approach the authors should at least clarify this point. Or they may consider omitting the last part of the analysis (validation of the predictive performance of clinical classifiers), which tends to undermine the premise of the work.

We apologise if the aim of this research was not made sufficiently clear for the reviewer. The objective of this research was not to develop predictive biomarkers of outcome but rather to identify molecular endophenotypes within the heterogeneous group of patients with IPAH. This would result in novel biological insights into disease mechanisms that allow better stratification for treatment choice, and highlight potential new drug targets. We had edited the introduction to make these aims more obvious.

“We investigated whether transcriptomic profiling of whole blood can provide more granular molecular ‘endophenotypes’ of H/IPAH to stratify patients better than is currently and whether it is permissible with the standard clinical classification. Furthermore, we hypothesised that these transcriptomic defined subgroups would provide novel insights into biological pathways driving disease, and potential novel drug targets.”

With regard to the last part of our analysis on an independent cohort, we included the definition of the clinical features that describe the transcriptomic subgroups so that other investigators could reproduce these subgroups based on clinical variables routinely collected in these patients, without the need to undertake transcriptomic analyses. This has been further strengthened by the addition of both internal and external validation of the relationships between the subgroups, transcripts and clinical features. We feel that our approach has robustly shown the existence of these subgroups. We have altered the final paragraph of the introduction to make this clearer.

“The gene expression profile of key cluster associated genes was subsequently confirmed, and the correlation with key clinical variables validated in both internal and external validation cohorts. This validates our approach and provides an alternative method to define these subgroups without the need for transcriptomic data.”

2. It is not apparent from reading the manuscript at first, but the authors are actually re-analyzing a dataset that they had previously generated and published results from using a different analytical workflow [ref 13]. Data reuse should definitely be encouraged, and it is not per se a negative point here. But some good practices would also need to be adopted. One would be to be upfront about it and clearly state that the data is being reused. A reason for that is that there also needs to be a discussion of the findings that have been reported earlier, and of the motivation of the re-analysis. What analysis gaps are being addressed? How do these findings and conclusions differ from those that were drawn earlier?

We appreciate that this is not a criticism but we did try to make this clear by referencing the manuscript referred to (Rhodes CJ, *et. al.* Whole-Blood RNA Profiles Associated with Pulmonary Arterial Hypertension and Clinical Outcome. *Am J Respir Crit Care Med.* 2020;202:586–594) in both the introduction and the methodology section. We agree that this is data reuse but this is not re-analysis, as different hypotheses are being addressed by each study.

In our earlier paper (Rhodes et al) we investigated a diagnostic signature and its prognostic utility from the whole blood transcriptome, across the whole cohort using supervised learning to compare healthy versus disease cases. The variation in whole blood transcriptome in this cohort and how that may result in subgroups (heterogeneity) of patients was not explored until now. In addition we expand on the cohort used in the Rhodes et al manuscript with the addition of clinical and RNA data from 197 PAH patients in the validation cohort. We have highlighted further these additional contributions in the Abstract and in the Introduction sections.

3. What compounds point #2 is that the authors had not made their data publicly available upon publication of their earlier work [ref 13], as is the custom and indeed a mandate in most quality journals. And they pass up the opportunity here once again, stating that data “are available from the corresponding author upon reasonable request”. Which practically is far from guaranteeing that the data will indeed be shared with the community for further re-use. Generating these data is obviously the culmination of many years of work and it is understandable that the authors wish to capitalize on those as much as possible. But after the publication of a secondary analysis, it seems that it would be indicated to make these data public and potentially further extending the impact of their work (and benefit to patients with this disease).

We agree with the reviewer about the requirement for data to be shared publicly and thank them for raising this concern. We edited our data availability section in the manuscript to include a link to a the European

Genome-phenome Archive (EGA) at the EMBL—European Bioinformatics Institute where the raw transcriptomic data can be downloaded. We are sure the reviewer appreciated the need for a controlled release of data to conform to the ethics under which the samples are collected given the sensitivity in human whole genome transcriptomics. The details on how to apply are included on both the EGA portal, the Shiny App, and below (via an email to cohortcoordination@medschl.cam.ac.uk).

Data Availability

The transcriptomic data from the PAH cases included in this manuscript and have been deposited in the European Genome-phenome Archive (EGA) at the EMBL—European Bioinformatics Institute under accession number EGADXXXXX. Application for access to the data from EGA can be made to the National Cohort of heritable and idiopathic PAH Governance committee by contacting cohortcoordination@medschl.cam.ac.uk. Furthermore, data used to generate statistics, plots and figures are accessible through our interactive portal found in <https://sheffield-university.shinyapps.io/ipah-rnaseq-app/>. Additionally, the code used to generate the results of this study is publicly available at https://github.com/BioSok/spectral_clustering_of_IPAH. Source data are provided with this paper.

4. An earlier study by a different group also associated the severity of IAH with high levels of expression of ALAS2 [ref 31]. This work would deserve to be discussed in more detail here (only one sentence as it is).

We thank the reviewer for pointing this out. We have expanded our description of this manuscript and included an edited section from the discussion below.

“Previous gene expression studies across multiple forms of PH, including IPAH significantly increased expression of ALAS2 in both systemic sclerosis-associated PAH (SSc-PAH) and IPAH³¹. In that study, for the IPAH patients ALAS2 also demonstrated strong correlation with right atrial pressure, pulmonary vascular resistance, pulmonary artery saturation and cardiac index³¹. These data, and our own observations (Figure 3) are further suggestive of a role for ALAS2, iron³² and hepcidin^{33,34} in pulmonary vascular remodelling, and PH.”

5. Addressing inter-individual variation employing unsupervised clustering approaches is suitable, although the approach has been employed for decades. Some novelty can be claimed concerning implementation itself, but dozens of clustering methods could be used with relatively similar outcomes. This is not a major criticism; it might just be indicated to downplay the novelty of the analytical approach accordingly.

We recognise the concern of the reviewer. Indeed there are an abundance of methods in the unsupervised learning field that are appropriate for certain data types. Most studies employ the widely-used clustering methods by default (e.g. hierarchical clustering) without utilising any kind of specific selection criteria for the chosen method to determine the suitability or effectiveness of the chosen methodology. In this study we benchmark three fundamentally different methods (hierarchical, k-means and spectral) and use partitioning consistency to determine which method produced the most stable and reproducible signal from our data type (RNA-sequencing). As shown in the supplementary figure 16 below, we have used 3 different kernels/distance measures for each clustering method and compared the agreement of partitioning on the same samples. Spectral clustering showed the highest consistency in detecting differences and subsequently partitioning patients in similar clusters independently of the kernel. Notable is the difference in intra-agreement of spectral (~75%) and k-means (-13%) clustering, which highlights the importance of the extra step of mapping data in a low dimensional space (as a similarity graph) in spectral clustering.

To make this distinction more clear we have included the following section in the supplementary material.

Clustering algorithms' partitioning consistency

In this study we compare three fundamentally different methods (hierarchical, k-means and spectral) and use partitioning consistency to determine which method picks up an underlying signal from our data type (RNA-sequencing). As shown in Supplementary figure 16, spectral clustering showed the highest consistency (Adjusted Rand Index) in detecting differences and subsequently partitioning patients in similar clusters independently of the kernel. Notable is the difference in intra-agreement of spectral (~75%) and k-means (~13%) clustering, which highlights the importance of the extra step of mapping data in a low dimensional space (as a similarity graph) in spectral clustering.

We also included the following text in section **Clustering algorithm selection** of the supplementary material: "There is a wealth of methods in the unsupervised learning field that are appropriate for certain data types. Most studies employ widely-used methods (e.g. hierarchical clustering) without utilising any kind of selection method that would point towards a certain effective methodology."

Furthermore, to highlight the point above we have included the following in the main manuscript in **Discussion**: "Most studies employ widely used clustering algorithms without exploring their data suitability while in this study we determine spectral clustering as the most consistent in detecting differences and subsequently partitioning RNA sequencing samples."

Supplementary Figure 16: The average adjusted rand index (ARI) of three clustering methods: spectral (blue line), hierarchical (green line) and k-means (red line) clustering. For each method 3 different distance measures/kernels were used and their ARI was averaged for each method.

Reviewer #2

The authors propose a machine-learning-based analysis of gene expression data for idiopathic pulmonary arterial hypertension classification. The authors' goal is to use a combination of unsupervised and supervised learning to identify subtypes of patients with different risk profiles and so survival rates.

Identification of the appropriate patient subgroups is an important problem. A novel, robust, and accurate method will be invaluable. Thus this manuscript addresses a critical challenge. The concept of this work is interesting, but there are a number of issues that need clarification.

Comments

1. How exactly an independent cohort was defined? Was it profiled in a different lab? Gene expression measurements are prone to batch-effect issues. I suggest the authors explore the performance of the selected subset of features/ signatures (clinical parameters+genes) and the model built on it on the completely independent dataset. GEO or ArrayExpress can be used for such purposes.

We agree that batch-effects are an important confounding variable in gene expression profiles. The samples and data used arise from the UK's National IPAH Cohort with samples collected from 8 sites across the UK (<https://ipahcohort.com>). We have included a supplementary PCA plot below to demonstrate that there are no obvious site specific effects. Furthermore, we have added 2 validation datasets in this revision as requested by the Editor and Reviews. These include validation of gene expression in an additional 197 patients from the National IPAH Cohort, and another validation cohort of 83 patients from the USA from which again measured the correlation between key genes in our RNA signature and the clinical variables. We hope that these two validation steps and the PCA analysis below (also included in the Supplementary Material) appease any concerns about batch specific findings.

Supplementary Figure 17 Plot of the first two principal components of the RNA-seq data derived from the 10,000 most variable genes according to IQR in our dataset. Each dot represents a distinct sample in the dataset coloured according to the institute that provided that sample. No discernible effect is seen due to Site.

Supplementary Figure 18 Boxplots showing the distribution of the first eight principal components of the RNA-seq dataset grouped according to the Site that provided the sample. No discernible effect is seen due to Site.

Internal validation of gene expression in clinical-feature defined subgroups

In order to validate the gene expression differences between subgroups in our validation cohort, TaqMan PCR was performed for 17 of the 27 genes previously associated with the subgroups and / or clinical variable correlations, with GAPDH used as the endogenous control gene. Nine of the 11 genes we measured demonstrated a fold-change between subgroup I and II in the same direction as the discovery cohort (Figure 6A). Differences in expression of key genes (IGHM, IGKV2.24, IGLV6.57 and NOG) were significant ($p < 0.01$) between subgroups I and II (Figure 6B).

Validation of gene and clinical feature correlations

The correlations between gene and clinical features observed in the discovery cohort were also examined in our validation cohort of 91 subjects, and also in an external cohort of 32 subjects with RNA collected from PBMCs (Romanoski et al. 2020). We found that 64 of the 90 (71%) correlations measured in these two independent cohorts were consistent with our discovery cohort (Supplementary Table 7).

2. HR and Cox regression were not adjusted to age, but the only gender. While it is seen that low-risk groups are somehow younger intermediate and high-risk groups and risk is growing with age. What was the age of healthy controls? Age is one of the important factors for survival in general. Could it be the case for the better survival of group II vs groups I and V? I suggest reporting sex and age-adjusted HR.

We apologise that this was not clearer in the manuscript. Data from healthy volunteers was not used in the cluster analysis, the focus was to describe subgroups of IPAH, not compare IPAH to healthy volunteers. To make this clearer we have included the following in the main manuscript in section **RNA data preprocessing** subheading of the **Methods**:

“The RNA sequencing and clinical data of healthy controls were not used in the main pipeline of this study. A secondary clustering with all patient and healthy samples was implemented to demonstrate the lack of pure patient and healthy subgroups within our cohort (Supplementary Figure 4B).”

We agree that age and sex are important covariates to include in our survival analysis, therefore, we included the following in the main manuscript in **Results** section **Unsupervised cluster analysis of whole-blood transcriptomes reveals five distinct subgroups of H/IPAH**:

“Age and sex were also included as covariates with the subgroups in a Cox regression model. Age above 52 years (median) was significantly associated with poor survival (HR=2.29) while gender showed no relationship with overall survival. Even with these covariates, the subgroup I was still significantly associated with survival (HR = 3.83) and was the biggest risk factor for poor outcome (**Supplementary Figure 14**).”

To address the second part of this suggestion we also included the following figure in the supplementary figures:

Supplementary Figure 14: Hazard Ratio of discovery cohort clustering adjusted for gender and age category of patients. Notation of (**) denotes a p-value of less than 0.01, (***) denotes a p-value of less than 0.001, while no stars denote non-significant p-values. Gender did not reveal any relationship with survival while an age over 52 was significantly associated with poor survival (HR=2.29). The most significant association with poor survival was found for patients participating in subgroup I (HR = 3.83).

3. Related to the previous point, it would be interesting to see how the standard clinical features would perform in risk prediction in comparison to gene expression profiling. Would it be possible to get similar clusters of patients with K-means/ Hierarchical clustering/Spectral clustering for their clinical features only?

We appreciate the proposition but as with previous response to Reviewer 1 we reiterate that the aim of this manuscript was to investigate the underlying biological heterogeneity in H/IPAH, with a view to better understanding the molecular mechanism and perhaps better stratify patients for specific therapies. We utilised the clinical variable to allow future / external identification of these specific subgroups without the need for whole blood transcriptomics. The clinical data also present some challenges technically for clustering algorithms given their different modalities and different distributions across variables.

Similarly, our aim was not to predict disease risk using gene expression, this would be a supervised classification and covered in our previous study (Rhodes et al. AJRCCM 2020). The aim was to uncover

novel RNA-based subgroups using unsupervised clustering. The fact that our transcriptomic subgroups provide prognostic information provide further validity that the underlying biological signal is important in the disease process. We we also saw that the subgroups were associated with different clinical risk scores ie. REVEAL 2.0. We also present some evidence that patients transition between these subgroups (Supplementary Figure 4) over time and with treatment but this requires further investigation with a longitudinal study. Because we used clinical outcome and REVEAL scores to validate our novel transcriptomic subgroups, we did not want to make a circular argument by combining the subgroups and REVEAL scores to predict outcome.

4. Names of packages used (for SVM, RF, and other methods and algorithms used) are lacking

We apologise for this omission. We have added the following to the Methods section **Identifying clinical signatures of subgroups:**

“The dataset was initially cleaned and filtered on 119 features that were identified by a domain expert from the original 887 features that described the dataset. Subsequently, any feature that had more than 5% missing data was dropped, and categorical features numerically encoded

All ML tasks were carried out using Scikit-learn ML framework version 0.23.2 in a Python 3 environment. As machine learning classifiers, we used Logistic Regression (LR), SVM, Random Forest (RF) and kNN. RF is a powerful ensemble learning technique especially for high dimensional classification tasks. LR was implemented using `sklearn.linear_model.LogisticRegression` using l2 penalty and default values used for all other parameters. SVM was implemented using `sklearn.svm.LinearSVC` with regularisation parameter C set to 1, and default values used for all other parameters. RF was implemented using `sklearn.ensemble.RandomForestClassifier` with default values used for all other parameters. kNN was implemented using `sklearn.neighbors.KNeighborsClassifier` with a number of neighbours (`n_neighbors`) set to 5 and default values used for all other parameters.

For feature selection tasks, we used ensemble feature selection based on recursive feature elimination (RFE) technique. RFE is a backward feature elimination technique that iteratively prunes the least informative feature(s) from a training dataset. A RFE based on a linear SVM starts by using all features to train an SVM model and ranks all features according to importance. The least ranked feature is removed from the training dataset and the SVM model refitted. This is iteratively done until only the required number of features remain. All features are also ranked according to importance.

Ensemble feature selection aggregates several feature rankings into a single consensus feature ranking to ensure robustness of the feature selection process and of selected features. Feature importance measures used for feature ranking are based on the hyperplane weight vector of a linear support vector machine (SVM). The weight vector quantifies the contribution of each feature to the construction of the hyperplane, and is used for ranking features according to importance.”

5. Samples are not defined - p 458. It is also confusing what is the difference between features and signatures. How many gene expression replicas were available per one patient?

In this manuscript, features are individual variables whereas signatures refer to the combination of variables, either genes or clinical variables. If there are specific examples where this needs further clarification, please let us know.

The whole blood RNAsequencing was performed on each patient individually. Five patients were sequenced twice across two sequencing runs to check that the expression profiles are reproducible, but only one replicate of each was used in the analysis. These replicate samples clustered together based on the principal components of their expression profiles. We added a new Supplementary Figure 20 to illustrate this.

Supplementary Figure 20: Principal components analysis of expression profiles from samples with a second replicate that was RNA sequenced (labeled as Trial2). Both replicates are clustered together according to the first four principal components.

6. It is not clear how authors moved from 25,955 genes to 1 to 20 features? What was the final set of genes used to trained supervised models?

We apologise for not making this clearer. We have added the following to the **Results** section to clarify this.

Unsupervised cluster analysis of whole-blood transcriptomes reveals five distinct subgroups of H/IPAH:

“Simultaneously, the 300 genes that produced the most stable expression data set (supplementary section Stability results) were utilised to identify unique subgroups of gene expression profiles, and describe the biological and clinical descriptors of these subgroups (**Figure 1**)”

We have also altered the following in section Spectral clustering: Gene expression subgroup identification of Methods:

“We performed cluster analysis to partition IPAH patients to distinct RNA based subgroups. The spectral clustering model (package kernlab v0.9-29) was selected as the most suitable unsupervised learning algorithm based on the highest partition consistency when comparing multiple algorithms. For the spectral clustering method, data points (i.e. patients) are embedded and partitioned in a low dimensional space in the form of a similarity graph, rather than being characterised by more than 25,000 gene dimensions. High partition consistency was defined as the high adjusted Rand Index (package fossil v0.3.7) and low standard deviation calculated between different variations of each clustering algorithm (k-means, spectral, hierarchical clustering), as described in Clustering algorithm selection. For the selection of the most appropriate clustering algorithm we utilised 25,955 genes across 359 IPAH patient samples (discovery cohort) after further filtering for repeated same-visit samples and non-H/IPAH diagnosis. To run the main spectral clustering partitioning we first selected the most relevant gene set, by ranking all genes based on the variability of their expression across all 359 patient samples using the stats v3.6.0 R package (supplementary section Feature selection of genes). Subsequently, several candidate gene sets of increasing size were

generated from the top ranking gene list, and the one that generated subgroups of highest stability was selected (package fpc v2.2-3, supplementary section Highest stability gene set). The number of IPAH subgroups was estimated through ensemble learning⁵⁶ utilizing 15 internal indexes calculated using the package dice R v0.6.0 (supplementary sections Optimal number of subgroups k and Internal Index Voting). The Radial Basis function kernel was used as the similarity measure with five target subgroups, identified as the optimal number of subgroups by an ensemble learning method. Further information on the selection of clustering algorithms and parameters can be found in the Supplementary Material.”

We used penalised regression to identify the genes associated with the RNA subgroups, but genes were not used in any of the supervised classification models. Instead, clinical signatures trained on the RNA subgroups were used for classification, as explained in section **Classification of new patients using signatures of Methods** in the main manuscript.

7. Figure 3 B: p-values are lacking for selected genes. Were observed differences significant? What was the cut-off for FC?

We apologise for this omission. In addition to adding the relevant significant difference to Figure 3B we have also included the following table in the supplementary as **Supplementary Table 8** to highlight all fold changes p values for genes.

We have also included the following in section **Gene signatures of subgroups** in the Supplementary material:

The p values of each gene when considering the fold change between subgroups I and II were calculated using a Welch Two Sample t-test on the raw values and presented in the **Supplementary Table 8**. The absolute lower cut off values of fold change (log2 scaled) is 0.28

8. Authors report that some of the features were imputed, but results are not reported. How many of the patients' profiles had missing values?

The information on missingness rates, patterns and causes are detailed in the **Missingness assessment and imputation** section of the **Methods** and in **Supplementary Figure 13**.

Supplementary Figure 13: A) Heatmap showing missingness across important clinical variables for the diagnostic dataset. B) Barchart showing the proportion of missing data and chart showing the combinations of missing data for the classifier variables from the diagnostic dataset. C) Heatmap showing missingness across important clinical variables for the cohort visit 1 dataset. D) Barchart showing the proportion of missing data and chart showing the combinations of missing data for the classifier variables from the cohort visit 1 dataset.

Reviewer #3

The investigators use clustering analyses and machine learning to examine biological and clinical heterogeneity in a cohort of patients with pulmonary arterial hypertension (PAH). They define 5 clusters on the basis of RNASeq data and move forward with describing 3 of the 5 clusters as the other two contain only 29 patients. They find that the 3 main clusters are associated with immunoglobulin levels and different gene expression patterns. The clusters are also associated with differences in their clinical risk as well as long-term outcomes. Focusing on clinical risk factors only, the clinical factors associated with each of the biological cohorts is validated in a second patient population. This is an interesting study of a relatively large PAH cohort.

1. More information is needed in this paper to understand the study cohort. Were hereditary patients genotyped? Were data from those patients who had a final diagnosis that wasn't PAH (supplementary figure 5) but did have RNASeq done taken out of all of the analyses?

We apologise if the cohort description is not clear. As discussed above in response to Reviewer 1 - point 2, the RNAseq data from the discovery cohort are the same as previously used to derive a diagnostic and IPAH signature and reported in Rhodes CJ, *et. al.* Am J Respir Crit Care Med. 2020;202:586–594. During the initial steps of this study, all patients with a non-PAH diagnosis were removed prior to any clustering procedure and therefore were not included in any of the subgroups or downstream analyses. The remaining 359 retained samples (with RNASeq data) were classified as IPAH/HPAH. At a later stage, a few patients were reclassified as other forms of PH during clinical follow up. These are observed in **Supplementary Figure 5** along with their subgroup membership.

Supplementary Figure 5: Revised diagnoses for each subgroup

As those samples were part of the main spectral clustering pipeline and distributed among all subgroups (no significant enrichment in any subgroup) they were retained and included in the results shown in the manuscript. The examination of our subgroup memberships and clinical signature did not show any clustering of those re-diagnosed samples, indicating that they did not bias the clustering. Additionally, the clustering of only IPAH patients from the final diagnosis produced very similar subgroups (differing survival - also see point 5 below) to the ones reported in the manuscript that includes both these re-classified patients, and the heritable PAH patients. We added analysis of IPAH-only samples in the Supplementary sections **Additional clustering pipeline of IPAH patients** and **Additional clustering pipeline of IPAH patients shows three survival groups**:

“To estimate the impact the 33 HPAH patients had on our main clustering pipeline, we examined their distribution across subgroups and ran an additional clustering pipeline including exclusively the 313 IPAH samples. As in the main pipeline, we utilised spectral clustering with the rbf kernel, the 300 most variable genes and identical preprocessing (section Sample and gene selection preprocessing)...

The 33 HPAH patients in our PAH cohort showed an equal distribution (~10%) among the subgroups of our initial clustering as demonstrated in Supplementary Table 9, indicating that they did not drive the clustering or greatly affected any subgroup characterisation. Furthermore, clustering the 313 IPAH samples resulted in 5 clusters whose survival curves are shown in Supplementary Figure 19. Similarly to the 3 major subgroups of the main clustering pipeline we can observe a group of patients with poorer survival over 5 years (clusters B and E, n=149), a group with higher survival (A and C, n=109) and a group with intermediate survival (D, n=55).”

The majority of included Heritable and Idiopathic PAH patients previously underwent whole genome sequencing and analysed for both rare (Graf et al Identification of rare sequence variation underlying heritable pulmonary arterial hypertension. *Nat Commun.* 2018;9:1416) and Common variants (Rhodes CJ, et al. Genetic determinants of risk in pulmonary arterial hypertension: international genome-wide association

studies and meta-analysis. *Lancet Respir Med.* 2019;7:227–238). The distribution of BMPR2 mutation, and the Sox-17 variant among the subgroups is shown in Figure 4C.

2. Taking out the transcripts that segregated patients by male sex is understandable based on data presented in the supplement, but since you are looking at RNA transcripts at a single point in time, it doesn't discount the possibilities that these are contributors to disease or resilience. This should be mentioned.

We thank the reviewer for raising an important point. To address this we altered/added the following text in **Discussion**:

“Although we cannot reject the possibilities of the aforementioned genes contributing towards PAH or resilience, we believe that their removal ensures that the clustering algorithm captures heterogeneity independent of sex associated expression variation. However, the interactions between gender and other autosomal genes in the context of PAH requires further study”

3. The paper focuses on 3 of the 5 cluster subgroups since 2 of the clusters contained only 29 patients. This raises the question about whether or not 5 is the optimal cluster number and whether or not it should have been 3. The various methods and measures of distance are shown in the supplement, but not all of these methods are not always appropriate for this type of data. By using each of these methods, did k=5 for all or is your internal indexes component of Fig. 1 meant to indicate that you took an average of the results of the other methods? If you perform clustering with k=3, what happens to the patients in subgroups 3 and 4 and does it change the transcripts associated with each subgroup? Since you did define 5 clusters, you should report on all of them throughout the Results section.

The reviewer raises an important point and one that we discussed at length at the outset of this study. To address this suggestion we included the following text in the supplementary section describing our approach for selecting k and hope that this provides enough information on the subgroup, how they were chosen, and their lineage.

“**Optimal number of subgroups k:** Determining the optimal number of clusters (k) is an inherently difficult task in unsupervised machine learning as it is always an educated estimation, since we do not know the actual number of categories within our data. Indeed some of the 14 used indexes are bound to not work on our data type (RNA-seq) and that is why we used an ensemble/voting method to estimate k (supplementary section **Internal index voting**) since we cannot base our estimation on any one index. The voting result (**Supplementary Table 1**) showed the clear majority of indexes to favour up to 5 clusters, with a preference to 2 clusters. The most important aspect in selecting the number of clusters in a data set is retaining as much information as possible, therefore selecting the highest supported k minimizes information lost. Following that notion, we retained the highest voted k = 5, where we discovered 3 distinct adequate sized clusters and 2 small clusters that, despite their interesting gene expression profiles (**Figure 2A**), were unable to show any statistical significance in follow-up work due to their small size. In **Supplementary Figure 15**, we demonstrate the flow of patients between clusterings along with the cluster sizes and the proportion/count of transferring patients across k. The colored nodes represent our subgroups I, II, III, IV and V. According to the clustering tree, the 3 main subgroups I, II and V remain clustered together when k = 2 (in a 341 sized cluster) and k = 3 (in a 295 sized cluster). This indicates that for k < 5 we are missing the information that separates these 3 distinct subgroups. The two smaller subgroups (III and IV) mostly originate from a group of patients (circled in green) that dissociates early on from main subgroups I, II and V implying that these samples show some differences even when less subgroups are requested. The remaining samples that end up in subgroup III have a common parent with subgroup II.”

Supplementary Figure 15: Clustree visualisation of the 5 subgroups (colored) discovered by our spectral clustering methodology. Edges represent the transfer of patients between clusterings of different k. Their opaqueness indicates the amount of patients that transferred.

We have also edited the following in section **Unsupervised cluster analysis of whole-blood transcriptomes reveals five distinct subgroups of H/IPAH** subheading of **Results:** “A clustering algorithm for selection and majority voting of multiple internal validation indexes (Supplementary Table 1) allowed us to identify as statistically optimal five distinct and stable subgroups of patients’ profiles (Figure 2A) while retaining the maximum heterogeneity information found in our data set.”

4. Use of the REVEAL 2.0 risk score is a nice way to describe the cohort. Does adding transcript profiles to this improve the predictive value (as the team has done in other publications)?

As highlighted in response to previous comments. Producing a better risk predictor was not an aim for this work. We have not therefore attempted to combine the transcriptomic or clinical features to the REVEAL risk score. We feel that this detracts from the main focus of this manuscript, which is to uncover novel RNA-based subgroups using unsupervised learning. The fact that our transcriptomic subgroups provide prognostic information provide further validity that the underlying biological signal is important in the disease process. We we also saw that the subgroups were associated with different clinical risk scores ie. REVEAL 2.0. We also present some evidence that patients transition between these subgroups (Supplementary Figure 4) over time and with treatment but this requires further investigation with a longitudinal study. Because we used clinical outcome and REVEAL scores to validate our novel transcriptomic subgroups, we did not want to make a circular argument by combining the subgroups and REVEAL scores to predict outcome.

5. The inclusion of hereditary and idiopathic PAH in the cohort is interesting, but begs the question about whether they should be analyzed separately. If you look at the idiopathic group alone, do you get the same results?

The reviewer raises an interesting point. Our approach to the study was that irrespective of genetic mutations, IPAH and HPAH share a common downstream biological processes but the route to these processes may be different within the subsets. Initially we just looked to see how patients with HPAH, or BMPR2 mutation, SOX-17 variant carriers were distributed across the subgroups. However to specifically address this comment we have performed this separate analysis as requested and include the following in the revised manuscript, in section **Unsupervised cluster analysis of whole-blood transcriptomes reveals five distinct subgroups of H/IPAH** of **Results**: **An additional clustering pipeline exclusively utilising 313 samples from patients with IPAH from our dataset (Supplementary section Additional clustering pipeline of IPAH patients)** also showed 5 subgroups, representing three significant corresponding survival profiles (Supplementary section **Additional clustering pipeline of IPAH patients shows three survival groups**), indicating that the inclusion of HPAH patients did not drive the partitioning procedure.

Furthermore, we included the following sections in the supplementary:

Additional clustering pipeline of IPAH patients: “To estimate the impact the 33 HPAH patients had on our main clustering pipeline, we examined their distribution across subgroups and ran an additional clustering pipeline including exclusively the 313 IPAH samples. As in the main pipeline, we utilised spectral clustering with the rbfdot kernel, the 300 most variable genes and identical preprocessing (section **Sample and gene selection preprocessing**).”

Additional clustering pipeline of IPAH patients shows three survival groups: “The 33 HPAH patients in our PAH cohort showed an equal distribution (~10%) among the subgroups of our initial clustering as demonstrated in **Supplementary Table 9**, indicating that they did not drive the clustering or greatly affected any subgroup characterisation. Furthermore, clustering the 313 IPAH samples resulted in 5 clusters whose survival curves are shown in **Supplementary Figure 19**. Similarly to the 3 major subgroups of the main clustering pipeline we can observe a group of patients with poorer survival over 5 years (clusters B and E, n=149), a group with higher survival (A and C, n=109) and a group with intermediate survival (D, n=55).”

Supplementary Table 9 | HPAH samples distribution across the 5 subgroups and their percentages against each subgroup total sample size

I	II	III	IV	V
12(9.3%)	6(5.35%)	3(15.7%)	1(10%)	11(12.3%)

Supplementary Figure 19: Survival of patients in clusters(A, B, C, D, E) created by clustering only 313 IPAH samples

6. The presence of the immunoglobulin chains is interesting and in line with prior reports about immune cell activation. Given that there are differences between immune and inflammatory cell numbers between the groups, if Ig transcripts are corrected for cell numbers, is there really still a difference between the groups to show actual differences?

We agree that the differential expression of the Ig genes across our novel subgroups is interesting, especially given the prior knowledge about immune response in PAH. Our results suggest that lymphocytes associated with these Ig genes were also in relative quantities across the subgroups. We have since validated the expression of the highly variable Ig genes (IGHV3, IGKV2, IGLV6) in two independent cohorts, including where RNA was extracted from peripheral blood mononuclear cells (PBMCs) instead of whole blood (new Figure 6 and Supplementary Table 7). This suggests that Ig transcripts may be more closely linked to immune cell numbers. We would also point to the strong role that the BMP antagonist Noggin plays in driving Subgroup II, perhaps suggesting that Noggin plays an important protective role in PAH. The aim of our study is to report the transcript changes associated with phenotypes, so whether we correct for cell numbers, or not, it does not alter that they are differentially expressed and reflect important differences in cell compositions. Correcting for cell numbers would not be possible at the moment as we do not have accurate estimates of Ig expression from each specific cell type in whole blood. Perhaps in the future when further data from single cell studies becomes available.

Figure 6 | Genes of interest with data based on our qPCR results of 91 patients (I = 53, II = 38) of the validation cohort. (A) Mean expression fold change (log₂ scaled) of the signature genes between validation subgroup I (immune inactive) and II (immune active). The fold ratio was generated based on negative delta Ct values (vs GAPDH). Genes over-expressed in subgroup I are denoted by light blue bars while genes primarily expressed in subgroup II are represented by dark blue bars. (B) The Relative Quantity (RQ) of each gene of interest relative to GAPDH with medians and significant differences shown. Notation of (**) denotes a p-value of less than 0.01, (***) denotes a p-value of less than 0.001.

7. On page 9, the statement that the average time between sampling and diagnosis was 5.3 years with a range that extends from 0.002-21.57 years is confusing. The way that this is written suggests that the samples were obtained first and the diagnosis of PAH was made later. Yet the sentence before it states that patients were diagnosed at a median of 45 years and sampled at a median of 52 years. Are you trying to indicate disease duration? If so, state accordingly. Also, the duration of time between diagnosis and sampling is vast, so please provide some more granular information on time from diagnosis to sampling. PAH patients with at least 22 years survival are quite different than patients with incident disease.

We apologise for the way this was written. This has now been changed to

Patients in this cohort were diagnosed at a median age of 45 years (IQR = 35-59 years) and sampled at a median age of 52 years (42-64) with an average of 5.3 years time between diagnosis and sampling.

The broad range in time from diagnosis to sampling reflects the nature of this cohort with a recruitment based on both prevalent and incident cases of H/IPAH. Below is the distribution of the time between diagnosis and sampling across the subgroups (<https://sheffield-university.shinyapps.io/ipah-rnaseq-app/>).

We completely agree that incident and prevalent patients are different, particularly those who have survived over 10 years. The aim of this work was to identify biologically similar subgroups of patients at the time of sampling regardless of how long they had been diagnosed to allow better stratification. We did also investigate in a small number of patients whether they would move between clusters overtime, partly to address the influence of disease progression and treatment on subgroup membership (Supplementary Figure 4).

8. Please also include age at diagnosis in Table 1 for each of the subgroups. Also, major characteristics should also include right heart cath data, RV function, and testing to rule out PE. PFT data alone are isolated descriptors. It also appears that no statistical testing took place.

Table 1 and Supplementary Table 2 have been updated to include age at diagnosis and clinical characteristics across RNA subgroups.

Table 1. Major clinical characteristics of the RNA subgroups in the discovery cohort (n = 359) at time of sampling.

	Low Risk Subgroup II (High immunoglobulin)	Intermediate Risk Subgroup V (Intermediate immunoglobulin)	High Risk Subgroup I (Low immunoglobulin)	All Patients
n	112	89	129	359
Age, years	46 [37-56]	52 [41-62]	57 [45-70]	52 [42-64]
Age at diagnosis, years	41 [31-51]	46 [37-55]	52 [42-67]	47 [35-59]
Gender: Female	82 (73%)	69 (78%)	80 (62%)	253 (70%)
Vasoresponse	10 (21.7%)	6 (13.6%)	6 (16.2%)	23 (16.4%)
Treatments				
Phosphodiesterase 5	12 (15.4%)	16 (21.9%)	22 (21.8%)	53 (19.4%)

Inhibitors (PDE5i)				
Endothelin Receptor Antagonist (ERA)	6 (7.69%)	13 (17.8%)	8 (7.92%)	33 (12.1%)
PDE5i & ERA Combination	42 (53.8%)	30 (41.1%)	53 (52.5%)	134 (49.1%)
Prostacyclin therapy	3 (3.85%)	1 (1.37%)	3 (2.97%)	7 (2.56%)
Calcium Channel Blockers	15 (19.2%)	13 (17.8%)	14 (13.9%)	45 (16.5%)
WHO Functional class				
I	18 (16.5%)	10 (11.2%)	6 (4.7%)	35 (9.8%)
II	45 (41.3%)	36 (40.4%)	44 (34.1%)	143 (40.2%)
III	43 (39.4%)	40 (44.9%)	65 (50.4%)	158 (44.4%)
IV	3 (2.8%)	3 (3.4%)	14 (10.9%)	20 (5.6%)
6 minute walking distance, m	397 [338-500]	420 [367-464]	327 [183-390]	387 [300-449]
NT-proBNP, ng/l	131.7 [54.5-362.0]	185.5 [76.3-463.5]	345.0 [91.0-1556.1]	222.5 [78.9-1162.8]
Forced Expiratory Volume [% predicted]	92 [82-101]	84 [72-98]	78 [66-98]	85 [68-100]
Forced Vital Capacity [% predicted]	101 (20)	99 (24)	93 (29)	97 (24)
Transfer factor of lung for carbon monoxide [% predicted]	93 [87-106]	97 [92-101]	88 [67-96]	94 [87-103]
Diagnostic Right Heart Catheter Study				
Mean pulmonary artery pressure, mmHg	47 [39-60]	52 [37-65]	56 [41-65]	51 [39-63]
Mean right atrial pressure, mmHg	8 [4-10]	8 [4-11]	11 [6-14]	9 [4-12]
Mean pulmonary Arterial wedge pressure, mmHg	10 [7-12]	10 [8-13]	12 [10-14]	11 [8-13]
Cardiac Index, L/min/m²	2.3 [1.6-2.8]	2.2 [1.7-2.4]	1.9 [1.5-2.5]	2.2 [1.6-2.5]
Pulmonary vascular resistance, Wood Units	8.1 [5.7-14.1]	15.0 [5.9-16.1]	8.4 [5.9-13.2]	8.9 [5.7-15.0]

Intervals describe first and third quartiles. Parenthesis describe standard deviation (SD).

9. More information is needed to understand how the variables were chosen for supervised machine learning. Was this by the investigators? Based on prior publications? What was significant in the dataset?

We apologise that this was not clearer. We have altered the section “**Identifying clinical signatures of subgroups**” in the **Methods** as follows:

The dataset was initially cleaned and filtered on 119 features that were identified by a domain expert from the original 887 features that described the dataset. Subsequently, any feature that had more than 5% missing data was dropped, and categorical features numerically encoded.

All ML tasks were carried out using Scikit-learn ML framework version 0.23.2 in a Python 3 environment. As machine learning classifiers, we used Logistic Regression (LR), SVM, Random Forest (RF) and kNN. RF is a powerful ensemble learning technique especially for high dimensional classification tasks. LR was implemented using `sklearn.linear_model.LogisticRegression` using l2 penalty and default values used for all other parameters. SVM was implemented using `sklearn.svm.LinearSVC` with regularisation parameter C set to 1, and default values used for all other parameters. RF was implemented using `sklearn.ensemble.RandomForestClassifier` with default values used for all other parameters. kNN was implemented using `sklearn.neighbors.KNeighborsClassifier` with a number of neighbours (`n_neighbors`) set to 5 and default values used for all other parameters.

For feature selection tasks, we used ensemble feature selection based on recursive feature elimination (RFE) technique. RFE is a backward feature elimination technique that iteratively prunes the least informative feature(s) from a training dataset. A RFE based on a linear SVM starts by using all features to train an SVM model and ranks all features according to importance. The least ranked feature is removed from the training dataset and the SVM model refitted. This is iteratively done until only the required number of features remain. All features are also ranked according to importance.

Ensemble feature selection aggregates several feature rankings into a single consensus feature ranking to ensure robustness of the feature selection process and of selected features. Feature importance measures used for feature ranking are based on the hyperplane weight vector of a linear support vector machine (SVM). The weight vector quantifies the contribution of each feature to the construction of the hyperplane, and is used for ranking features according to importance.

10. CRP shows up in all your groups and BMI in several. You don't indicate if the levels are the same for the groups, high vs. low, etc. The levels aren't reported. It's well recognized that there are age, race/ethnicity, and sex differences in values of the factors. How did you adjust for this?

For the levels of various clinical variables (including CRP and BMI), we have reported their distribution of values across subgroups in **Figure 5** (for the discovery cohort) and **Supplementary Figure 10**. The discovery cohort was clustered solely based on RNA profiles thus our algorithm remaining agnostic to any of the clinical variables. The subgroup classifier that was tested on the validation cohort was composed of clinical variables with the highest contribution towards separating the RNA subgroups. This included both CRP and BMI in the case of our univariate feature selection model and only BMI in the case of our multivariate feature selection model (**Figure 5**). We checked that the subgroups were independent of age and ethnicity differences (**new Supplementary Figure 14**) and our cohort was primarily caucasian.

Supplementary Figure 14: Hazard Ratio of discovery cohort clustering adjusted for gender and age category of patients. Notation of (**) denotes a p-value of less than 0.01, (***) denotes a p-value of less than 0.001, while no stars denote non-significant p-values. Gender did not reveal any relationship with survival while an age over 52 was significantly associated with poor survival (HR=2.29). The most significant association with poor survival was found for patients

11. How many patients were on disease-modifying drugs, steroids, or anti-inflammatory agents at the time of sampling? Since this is a PAH cohort, it is expected that the CTD patients at a minimum would fall into this category.

We have highlighted “Treatments” in **Table 1** where PAH-specific therapies and calcium channel blockers are reported for the main 3 clusters. As described this is a cohort of heritable and idiopathic PAH, no CTD patients were included.

12. The clinical features list should be put in table format to make it easier to understand the differences between the groups.

We apologise for this omission. The data were in the Supplementary material but not referenced within the main text of the manuscript. To address this we have added the following text in the Results section titled

“Clinical signatures describe RNA-based subgroups and are associated with subgroup-specific genes: ...and in table format in (Supplementary Figure 2).”

13. The validation cohort study is perplexing since you use RNASeq to identify clinical variables to predict outcomes and then validate the clinical variables. One could make the argument that you could just use the REVEAL risk score to predict outcomes. Is the discriminatory power of the variables identified by your methodology better than the REVEAL risk score in the validation cohort? If the idea is also to identify targets for therapies (last sentence of your abstract), then how does validating clinical characteristics in a separate cohort help this cause? It seems that you would have to show that the validation cohort has relatively the same RNASeq profile.

We thank the reviewer for this comment but again we would stress that we were not aiming to develop another tool for PAH risk prediction. We identified the clinical variables that defined the transcriptomic subgroups so that we could validate whether these could be used to replicate the subgroups in other clinical cohorts, which we did. This was designed to show that the subgroups we describe are robust and could be associated with biological mechanisms, or clinical outcomes.

To address the concern raised here and emphasised by the Editor we have undertaken two additional steps to validate our findings. Firstly we obtained stored TEMPUS RNA samples from the national biorepository in Cambridge for the patients included in the the validation cohort who had not undergone RNA sequencing. We extracted these samples and ran targeted TaqMAN quantitative PCR analysis of key genes within the subgroups to demonstrate that the gene expression within these clinical variable defined subgroups was equivalent to the discovery subgroups defined by gene expression. We also examined the correlation between these genes and the clinical variables to confirm their association (new section **Clinical variable and gene correlations**). Secondly, we have obtained data from an external cohort with RNA sequencing through collaboration with Dr Ankit Desai at Indiana University to provide external validation for the relationship between these clinical variables and subgroup associated genes.

We have added this data to the main manuscript including two new sections “**Internal validation of gene expression in clinical-feature defined subgroups**” and “**Validation of gene and clinical feature correlations**” in the **Results**, a new **Figure 6**, and **Supplementary Table 7**.

As mentioned above we also added the new section **Clinical variable and gene correlations** in **Methods**. We calculated correlations between the clinical and gene signatures we generated in previous steps of this study. For discovery and validation cohorts we used the `rcorr` function of R package `Hmisc` (version 4.5-0). For the external validation we used the values found in ⁶⁴.

Internal validation of gene expression in clinical-feature defined subgroups

In order to validate the gene expression differences between subgroups in our validation cohort, TaqMan PCR was performed for 17 of the 27 genes previously associated with the subgroups and / or clinical variable correlations, with GAPDH used as the endogenous control gene. Nine of the 11 genes we measured demonstrated a fold-change between subgroup I and II in the same direction as the discovery cohort (Figure 6A). Differences in expression of key genes (IGHM, IGKV2.24, IGLV6.57 and NOG) were significant ($p < 0.01$) between subgroups I and II (Figure 6B).

Validation of gene and clinical feature correlations

The correlations between gene and clinical features observed in the discovery cohort were also examined in our validation cohort of 91 subjects, and also in an external cohort of 32 subjects with RNA collected from

PBMCs²⁷. We found that 64 of the 90 (71%) correlations measured in these two independent cohorts were consistent with our discovery cohort (Supplementary Table 7).

Figure 6 | Genes of interest with data based on our qPCR results of 91 patients (I = 53, II = 38) of the validation cohort. (A) Mean expression fold change (log₂ scaled) of the signature genes between validation subgroup I (immune inactive) and II (immune active). The fold ratio was generated based on negative delta Ct values (vs GAPDH). Genes over-expressed in subgroup I are denoted by light blue bars while genes primarily expressed in subgroup II are represented by dark blue bars. (B) The Relative Quantity (RQ) of each gene of interest relative to GAPDH with medians and significant differences shown. Notation of (**) denotes a p-value of less than 0.01, (***) denotes a p-value of less than 0.001.

Supplementary Table 7 | Correlations between discovery, validation and external validation cohorts. In the discovery set Spearman correlations were calculated between RNA-seq TPM values, the validation set between negative delta Cts values with GAPDH used as the endogenous control gene. Green cells denote agreement in gene-clinical variable correlation directionality between the cell's data set and the discovery data set. Red cells denote disagreement/opposite correlation between the two data sets. Notation of (**) denotes a p-value of less than 0.01, (***) denotes a p-value of less than 0.001, while no stars denote non-significant p-values.

Genes	Discovery [n= 359]			Validation [n =91]			External Validation [n =32]		
	Age	BMI	SixMWD	Age	BMI	SixMWD	Age	BMI	SixMWD
ALAS2	0.2***	0.38***	-0.32***	-0.006	0.06	-0.1	-0.08	0.31	-0.35
C4BPA	0.01	0.02	-0.07	0.14	-0.11	-0.13	0.02	-0.19	0.25

CRISP3	0.16*	0.14	-0.16*	0.11	0.016	-0.22	-0.07	0.14	0.05
CTSG	0.19***	0.14*	-0.12*	0.07	0.06	-0.17	0.06	0.19	-0.12
IFI27	0.19**	0.16*	-0.22***	0.24*	0.004	-0.28*	0.1	0.1	-0.32
IGHM	-0.29***	-0.18***	0.2***	-0.42***	-0.21*	0.18	-0.15	0.33	-0.02
IGHV3.48	-0.27***	-0.08	0.14*	-0.31	0.34	-0.5*	-0.11	0.22	0.12
IGKV2.24	-0.25***	-0.26***	0.24***	-0.45***	-0.17	0.11	-0.07	0.14	0.15
IGLV6.57	-0.21***	-0.16*	0.15*	-0.5***	-0.08	0.16	-0.31	0.32	0.04
LTF	0.18**	0.19***	-0.13*	0.08	0.12	-0.23*	0.01	0.08	0.01
NEBL	0.01	0.05	0.01	-0.09	-0.01	-0.23	0.03	0.12	-0.16
NOG	-0.44***	-0.19***	0.2***	-0.58***	-0.2*	0.18	-0.13	0.02	0.14
NPRL3	0.05	0.18*	-0.13*	-0.002	-0.09	-0.1	0.03	-0.06	-0.41
PI3	0.15**	0.25***	-0.17**	0.1	0.04	-0.21	-0.13	-0.02	0.31
SMIM11A	-0.03	-0.2***	0.12*	-0.24*	-0.13	0.06	0.1	-0.04	0.18

Additionally, we added the section “qPCR on validation cohort” in **Methods**:

Frozen Tempus tubes collected from patients in the validation cohort, collected under the UK National Cohort study were obtained, RNA was extracted using Maxwell® 16 LEV simplyRNA Blood Kit (Cat.# AS1310) as described in manufacturers instructions on the Maxwell® 16 Instrument (Cat.# AS2000) . Extracted RNA was transcribed using the High-Capacity-RNA-to-cDNA kit (Thermo Fisher Cat.# 437406) following manufacturers instructions. Resultant cDNA was analysed using custom taqman array cards (Thermo Fisher Cat.#4342249) with Fast Advanced Mastermix (Thermo Fisher Cat.# 4444964), Samples were run 8 to a card across 25 cards with 24 primer probes per sample (18S-Hs99999901_s1, ACTB-Hs00357333_g1, ALAS2-Hs01085701_m1, BMP2-Hs00176148_m1, C4BPA-Hs00426339_m1, CRISP3-Hs00195988_m1, CTSG-Hs00175195_m1, GAPDH-Hs02786624_g1, HPRT1-Hs02800695_m1, IFI27-Hs01086373_g1, IGHM-Hs00941538_g1, IGHV3-75-Hs03832008_sH, IGKV2-24-Hs06671746_g1, IGLV6-57-Hs01696637_s1, LINC00221-Hs01382601_m1, LTF-Hs00914334_m1, MT-RNR1-Hs02596859_g1, NEBL-Hs01067284_m1, NOG-Hs00271352_s1, NOS2-Hs01075529_m1, NPRL3-Hs00429221_m1, PI3-Hs00160066_m1, SMIM11A;SMIM11B-Hs00938773_m1, XIST-Hs01079824_m1. These assays were performed in duplicate using the Applied Biosystems 7900HT Fast real-time PCR system with the TaqMan Low Density Array card block following calibration using the TaqMan Low Density Array Calibration Kit (Thermo Fisher Cat.# 10341465). Ct values were determined with Automatic thresholding in the SDS2.4 software. GAPDH- Hs02786624_g1 was used as a control. Relative quantity was calculated using the $\Delta\Delta C_t$ method.”

REVIEWERS' COMMENTS

Reviewer #1 (Remarks to the Author):

The revisions made to the manuscript and responses from the reviewers are acceptable from my end.

It is still somewhat disappointing that the data is not being shared openly.

The number of patient RNAseq profiles openly shared in GEO reach easily in the tens if not hundreds of thousands.

[https://www.ncbi.nlm.nih.gov/gds?term=\("high throughput sequencing"%5BPlatform Technology Type%5D\) AND homo sapiens%5BOrganism%5D](https://www.ncbi.nlm.nih.gov/gds?term=()

If there are any concerns that sequence data could be identifiable, raw counts and normalized counts could also be shared. This is not sensitive information.

The authors could also still chose to make explicit in the manuscript the fact that at least one part of the data was reused from an earlier study and explain what the differences were in terms of the aims & approaches.

Reviewer #2 (Remarks to the Author):

First, I thank the authors for extensive revisions, and, to all appearances, having taken the comments of all reviewers seriously. The authors have substantially improved the manuscript by including the more comprehensive mortality analysis, describing methods more thoroughly, however, there are still several areas needing further attention.

Method section requires some work on clarity to be suited to the general audience of Nature Communications, for example, I believe there is no need to use the notion of parameters used in code in the main text (ex. `n_neighbors` and etc). And I suggest moving some of the details to the Supplementary section.

Thanks for clarifying the number of replicas. To fully utilize the data produced, I believe It would be really interesting to see would the endophenotype classification be preserved across replicas. By using the trained model, authors can simply predict those classes on replicas that were excluded from the analysis.

The figure legends seem to be omitted from the main text.

The code repository does seem to contain only R code, no Python code for ML models was provided. As most of the models are initialization dependant, it is important to provide code to ensure reproducibility of the results.

Reviewer #3 (Remarks to the Author):

The authors have been quite responsive in their revision and this is much appreciated by this reviewer.

Two quick points requiring clarification:

1. A description of the outside validation cohort in the Methods section isn't seen.
2. Since the validation cohort didn't have RNASeq data and gene expression was confirmed by PCR using a subset of genes (9 of 17), stating that this validated the discovery cohort is an overstep and should be toned down.

Reviewer #1:

The revisions made to the manuscript and responses from the reviewers are acceptable from my end.

1. It is still somewhat disappointing that the data is not being shared openly. The number of patient RNAseq profiles openly shared in GEO reach easily in the tens if not hundreds of thousands. [https://www.ncbi.nlm.nih.gov/gds?term=\(%5BPlatform+Technology+Type%5D\)+AND+homo+sapiens%5BOrganism%5D](https://www.ncbi.nlm.nih.gov/gds?term=(%5BPlatform+Technology+Type%5D)+AND+homo+sapiens%5BOrganism%5D). If there are any concerns that sequence data could be identifiable, raw counts and normalized counts could also be shared. This is not sensitive information.

Answer

We have updated the data availability statement in order to include 1) a statement of the reasons for controlled access (eg., privacy, ethical/legal issues) 2) a detailed description of the conditions of access including contact details for access requests 3) timeframe for response to requests 4) restrictions imposed on data use via data use agreements. Since this is a genotype-phenotype study, uploading RNA count data to GEO without the clinical information (which is sensitive) would not be useful to any researchers, and this is why we deposited it with EGA instead. Please note that we also provide a publicly accessible (after registration) shiny app for readers to view gene level expression data.

2. The authors could also still chose to make explicit in the manuscript the fact that at least one part of the data was reused from an earlier study and explain what the differences were in terms of the aims & approaches.

Answer

We have further edited the introduction to make this point clear.

Reviewer #2:

First, I thank the authors for extensive revisions, and, to all appearances, having taken the comments of all reviewers seriously. The authors have substantially improved the manuscript by including the more comprehensive mortality analysis, describing methods more thoroughly, however, there are still several areas needing further attention.

1. Method section requires some work on clarity to be suited to the general audience of Nature Communications, for example, I believe there is no need to use the notion of parameters used in code in the main text (ex. `n_neighbors` and etc). And I suggest moving some of the details to the Supplementary section.

Answer

We tried to provide an in depth description of methods to enable reproducibility, but have now moved some finer details about model training and optimisation to the Supplementary.

2. Thanks for clarifying the number of replicas. To fully utilize the data produced, I believe It would be really interesting to see would the endophenotype classification be preserved across replicas. By using the trained model, authors can simply predict those classes on replicas that were excluded from the analysis.

Answer

The few replicas were performed on RNA extracted from the same blood sample to conduct quality control of our RNA sequencing. It would be interesting to take additional blood samples from the same patient especially since we observed some patients with multiple samples changing endophenotypes over time. We are planning a larger study to look at longitudinal changes.

3. The figure legends seem to be omitted from the main text.

Answer

We apologise for the formatting issue and have updated the figure legends.

4. The code repository does seem to contain only R code, no Python code for ML models was provided. As most of the models are initialization dependant, it is important to provide code to ensure reproducibility of the results.

Answer

Full R and python code is now provided in a public code repository at <https://zenodo.org/badge/latestdoi/299615578>.

Reviewer #3:

The authors have been quite responsive in their revision and this is much appreciated by this reviewer.

Two quick points requiring clarification:

1. A description of the outside validation cohort in the Methods section isn't seen.

Answer

We have now added this to the Methods.

2. Since the validation cohort didn't have RNASeq data and gene expression was confirmed by PCR using a subset of genes (9 of 17), stating that this validated the discovery cohort is an overstep and should be toned down.

Answer

We have now toned down the validation section that used PCR.